# Fluorescent tracking identifies key migratory dendritic cells in the lymph node after radiotherapy

Tiffany C Blair[1], Shelly Bambina[1], Gwen F Kramer[1], Alexa K Dowdell[1], Alejandro F Alice[1], Jason R Baird[1], Amanda W Lund[2], Brian D Piening[1], Marka R Crittenden[1,3], Michael J Gough[1]

**Radiation therapy generates extensive cancer cell death capable of promoting tumor-specific immunity. Within the tumor, conventional dendritic cells (cDCs) are known to carry tumor-associated antigens to the draining lymph node (TdLN) where they initiate T-cell priming. How radiation influences cDC migration is poorly understood. Here, we show that immunological efficacy of radiation therapy is dependent on cDC migration in radioimmunogenic tumors. Using photoconvertible mice, we demonstrate that radiation impairs cDC migration to the TdLN in poorly radioimmunogenic tumors. Comparative transcriptional analysis revealed that cDCs in radioimmunogenic tumors express genes associated with activation of endogenous adjuvant signaling pathways when compared with poorly radioimmunogenic tumors. Moreover, an exogenous adjuvant combined with radiation increased the number of migrating cDCs in these poorly radioimmunogenic tumors. Taken together, our data demonstrate that cDC migration play a critical role in the response to radiation therapy.**

## Introduction

Radiation therapy has been used for over a century to treat malignancies, but the mechanisms underlying its impact on the immune system are not fully understood. It is well established that treatment with radiation induces lethal DNA damage in cancer cells, resulting in widespread cancer cell death (1, 2, 3). Recent evidence has demonstrated that this cell death is capable of interacting with the immune system to promote cancer-specific immunity (4, 5). CD8[+] T cells are known to play a critical role in controlling tumors after radiation therapy, and therefore, it is logical that the DCs that prime these tumor-reactive T cells are critical for the response to radiation (4). It has been proposed that the release of tumor-associated antigens and endogenous adjuvant signals from dying cancer cells after radiation therapy can function to activate DCs and thus serve as an in situ vaccine against

tumors. However, recent data demonstrate that tumor control by radiation therapy and checkpoint inhibitor immunotherapy is dependent on preexisting T-cell responses (6), so the role of DCs in priming new T cells after radiation therapy is unclear.

Priming of tumor-specific CD8[+] T cells requires specialized DC subsets with the capacity to cross-present tumor-associated antigens. Conventional type 1 DCs (cDC1s) have been shown to excel at cross-presenting antigens to CD8[+] T cells (7, 8, 9). In murine models, cDC1s are defined by their expression of the transcription factors ID2, IRF8, ZBTB46, and BATF3 (10). cDC1s can be further divided into those with the capacity to migrate from the tissue to lymphoid organs (migratory CD103[+] cDC1s) and those that remain resident to lymphoid organs (resident CD8$\alpha$[+] cDC1s) (9, 11). Migratory CD103[+] cDC1s are one of the primary cell types capable of trafficking intact tumor-associated antigens from the tumor to the TdLN via a CCR7-dependent mechanism to initiate CD8[+] T-cell priming (12). The effect of radiation therapy on the different DC subsets and their importance in tumor-specific T-cell priming remains to be determined.

Radiation therapy has been reported to drive the release of adjuvant-like compounds capable of inducing DC maturation (5, 13). In preclinical models, cDC1s have been shown to support antitumor immunity because Batf3[−/−] mice that lack cross-presenting cDC1s fail to reject highly immunogenic tumors (7). The efficacy of radiation therapy is significantly diminished in Batf3[−/−] mice (14, 15). However, in these studies, cDC1s were depleted in the animals throughout tumor development, making it difficult to determine whether cDC1s contribute to the initial priming of tumor-reactive T cell at tumor implantation or whether they function to prime new T cells after treatment with radiation. In radioimmunogenic tumors, which depend on the adaptive immune system for their enhanced response to radiation (16), we have demonstrated that radiation drives intratumoral cDC1 maturation (17). We also demonstrated that this process failed to occur in tumors that are poorly radioimmunogenic and do not generate antitumor immunity after radiation (17). Ectopic delivery of innate adjuvants can restore maturation of DCs and improve tumor control with radiation therapy in a DC-dependent manner (18). Maturation and migration are linked properties of DCs that permits cross-presentation of tumor-associated antigen in the TdLN, and cDC1 migration to the

[1]Earle A Chiles Research Institute, Robert W Franz Cancer Center, Providence Portland Medical Center, Portland, OR, USA [2]Ronald O Perelman Department of Dermatology, Department of Pathology, NYU Grossman School of Medicine, New York, NY, USA [3]The Oregon Clinic, Portland, OR, USA

Correspondence: michael.gough@providence.org

TdLN is likely required to initiate priming of new tumor-reactive T cells that cannot access peripheral tissues (19). However, given that radiation up-regulates direct antigen presentation by cancer cells in the tumor environment (20, 21) and that tumor control by radiation therapy and immunotherapy can function independently of T cell recirculation out of lymph nodes (6, 22, 23), it remains unproven whether cross-presentation by DCs occurs in the tumor or the TdLN after radiation.

In this study, we aimed to understand how DC migration contributes to the responsiveness of tumors to radiation therapy. We address this using site-specific fluorescence tagging of tumor DCs, mixed bone marrow chimeras with selective conventional dendritic cell (cDC) depletion to identify mechanisms of migration, and single-cell RNA sequencing to identify mechanisms regulating maturation after radiation therapy. Using radioimmunogenic tumors that are known to depend on the immune system for improved tumor control after radiation therapy, we demonstrate that cDC migration to the TdLN is required for tumor control and regression after treatment. We demonstrate in radioimmunogenic tumors that radiation does not change the number of cDC1s migrating from the tumor to the TdLN but instead increases the expression of activation markers on tumor-migratory cDC1s. By contrast, in poorly radioimmunogenic tumors, we have shown that radiation decreases the number of tumor-migratory cDC1s in the TdLN. Transcriptional analysis revealed that DCs from radioimmunogenic tumors express genes associated with activation by innate endogenous adjuvants, and co-administering radiation with exogenous adjuvants overcomes the failure for DCs to migrate in poorly radioimmunogenic tumors. Taken together, these data demonstrate that DC migration from the tumor to the TdLN is critical to generating a productive tumor-specific immune response after radiation therapy.

## Results

### DCs photoconverted in the tumor are found in the TdLN as migratory CD103⁺ cDC1 and CD11b⁺ cDC2 subsets

To first define the cDCs that migrate to the TdLN after radiation therapy, we set out to understand the kinetics and phenotype of cDCs migrating directly from the tumor to the TdLN. In our models, the inguinal LN drains the tumor along with other surrounding tissues and as a result, makes it difficult to assess changes in only tumor-migratory DC populations after treatment (24, 25). To overcome this issue, we used Kaede photoconvertible mice, which express the Kaede-green fluorescent protein that can be converted into the Kaede-red fluorescent protein upon exposure to violet light (26). Radioimmunogenic MC38 tumors were implanted into Kaede mice, and when tumors reached an average diameter of 5–6 mm, the animals were covered in aluminum foil. Only the tumor was left uncovered and exposed to a 405-nm LED light source for 5 min to photoconvert infiltrating cells in the tumor (17). The tumor was then treated with 12 Gy radiation (Fig 1A). The TdLN was harvested at 1, 2, and 3 d post-photoconversion for analysis by flow cytometry (Fig 1A). Our analysis revealed that the most of the converted (Kaede-red⁺) cells in the TdLN were T cells (data not shown) and DC

populations, as has been previously published (Figs 1B and S1) (27, 28). Strikingly, we noted that within the TdLN, most of the converted Kaede-red⁺ DCs were migratory CD103⁺ cDC1 and CD11b⁺ cDC2 subsets, validating that these were indeed migratory populations (Fig 1B). Further confirming the specificity of this model, we did not find any converted DCs in the contralateral inguinal LN (Fig 1B). Given that the overlaying skin is included in the radiation field and the field of UV conversion, it is possible that alternative DC populations such as Langerhans cells are among the converted cells in the TDLN. However, Langerhans cells in the lymph node are defined as F4/80⁺CD11b⁺CD103⁻ (29, 30); therefore, these are not among the CD103⁺ DC population that differentially matures after radiation therapy. To compare the effect of photoconversion of skin in the absence of a tumor, we analyzed the lymph node draining unconverted skin versus photoconverted skin. One day after conversion, we find that as with the tumor draining lymph node, we only see significant proportions of converted cells in the migratory CD103⁺ cDC1 and CD11b⁺ cDC2 subsets (Fig S2). To address the extent of photoconversion in the tumor, we analyzed the photoconversion of myeloid populations in the tumor immediately after photoconversion and over time, in the presence and absence of radiation therapy. We demonstrate that up to 80% of myeloid cells in the tumor are photoconverted immediately after UV exposure (Fig S3). When we follow photoconversion over time, we find that more stable populations such as F4/80⁺ Ly6C⁻ MHC-II⁺ TAM retain a significant converted population 3 d after conversion (Fig S3). Neutrophils, which are very short-lived, show a steady loss of converted cells in the tumor. Interestingly, monocytes, which are likely to differentiate into different cell types have an extremely short period of conversion (Fig S3) and are likely replaced rapidly by newly recruited monocytes that are not converted. The DC populations show a decline in converted cells over time, more similar to neutrophils than macrophages or monocytes. In each cell type, we cannot detect a consistent effect of radiation therapy on the proportion of converted cells in the tumor (Fig S3), suggesting that radiation is not directly impacting turnover of the Kaede protein in the tumor myeloid populations. Taken together, these data suggest that the Kaede mice are a useful tool to study DC populations in the TdLN that have directly migrated from the conversion field after treatment with radiation therapy.

### Radiation increases the proportion of activated tumor-migratory cDCs in the TdLN of radioimmunogenic tumors

We investigated two major mechanisms by which radiation could influence cDC migration: (1) increasing the number of migratory cDCs after treatment or (2) changing the phenotype of these cells after therapy. Using the Kaede mice, we monitored the proportion of converted Kaede-red within each DC subset in the TdLN over time and noted that radiation did not alter to proportion of converted DCs in the TdLN when compared with untreated controls (Fig 1C i–iv). However, we did observe that in each group, the proportion of converted DCs declined over time for both CD103⁺ DCs and CD11b⁺ DCs (Fig 1C i and ii). To further assess trafficking DCs, we specifically gated on these tumor-migratory populations to evaluate changes in their absolute numbers and phenotype after radiation. Radiation did not change the total number of converted (Kaede-red) CD103⁺

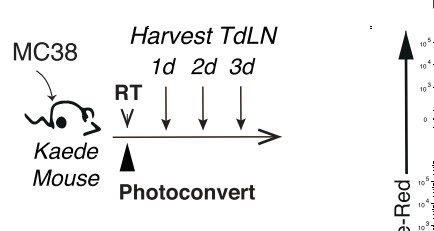

A) Experiment Design

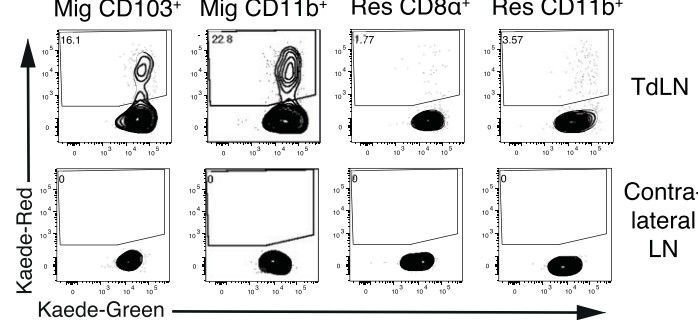

B) Converted (Kaede-Red) cDC Subsets - 1d Post

C) Frequency of Converted (Kaede-Red) cDC Subsets in the TdLN

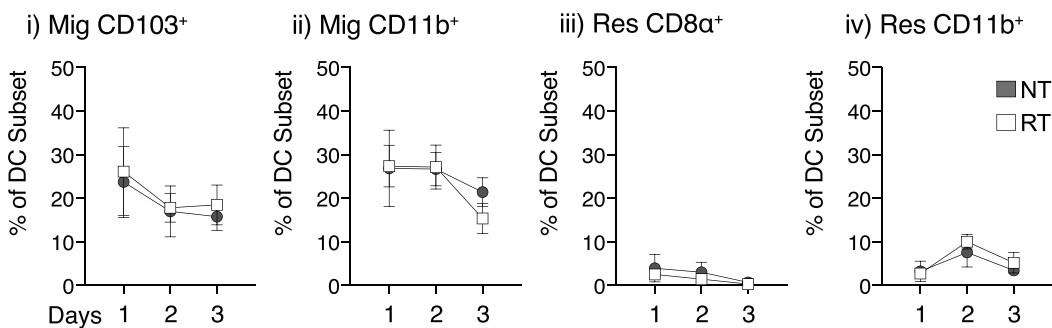

D) Day 3 converted (Kaede-Red) CD103⁺ cDC1s    E) Day3 converted (Kaede-Red) CD11b⁺ cDC2s

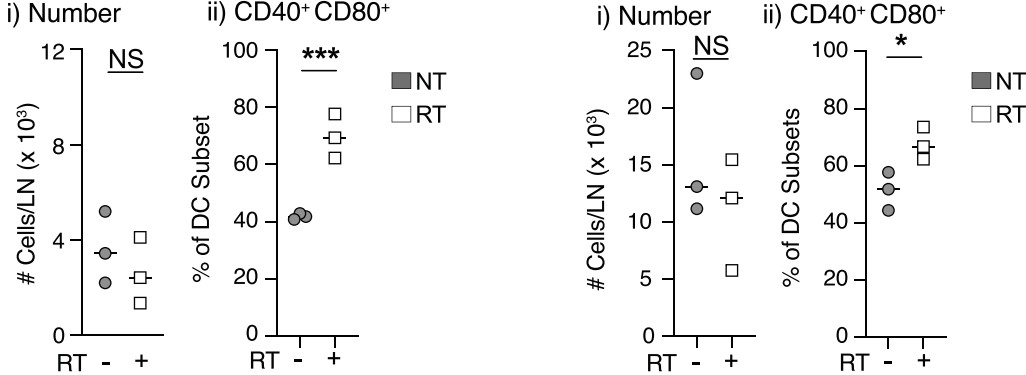

**Figure 1.   Increased proportion of tumor-migratory DCs co-expressing CD40 and CD80 in the TdLN of radioimmunogenic tumor bearing mice after radiation therapy.**
**(A)** Experiment design: MC38 tumors were established in Kaede mice. When tumors reached an average of 5–6 mm average diameter, they were photoconverted and then treated ±12 Gy CT-guided radiation therapy, and the TdLN was harvested for analysis by flow cytometry at indicated timepoints. **(B)** Representative flow plot from 1 d post-photoconversion for migratory (mig) and resident (res) DC populations in the TdLN or contralateral inguinal LN showing converted (Kaede-red⁺) DCs. **(C)** For untreated (NT, gray) or radiation (RT, white) treatment, the proportion of Kaede-red⁺ DCs within each DC subset in the TdLN: (i) Mig CD103⁺, (ii) Mig CD11b⁺, (iii) Res CD8α⁺, and (iv) Res CD11b⁺. **(D)** (i) The absolute number of Kaede-red⁺ CD103⁺ DC/dLN and (ii) frequency of CD40⁺ CD80⁺ within converted CD103⁺ DCs for NT (gray) or RT (white) groups at d3 after conversion. **(E)** (i) The absolute number of Kaede-red⁺ CD11b⁺ DC/dLN and (ii) frequency of CD40⁺ CD80⁺ within converted CD103⁺ DCs for NT (gray) or RT (white) groups at d3 after conversion. Data represent the mean ± SEM of each group. Results shown are representative of two independent experiments with n = 3–6 animals/group. NS = not significant, *P < 0.05, and ***P < 0.001.

cDC1s or CD11b⁺ DCs when compared with untreated controls (Fig 1D i and E i). However, the number of converted CD103⁺ cDC1s and CD11b⁺ DCs slowly declined in both groups (Fig S4). Thus, radiation does not appear to change the kinetics of DC migration from the tumor to the TdLN. The next question was whether treatment

impacted the phenotype of DC populations migrating from the tumor. The co-stimulatory molecules CD40 and CD80 are up-regulated during DC maturation and are important for T-cell priming (31). Starting 2 d after treatment, there was a significant increase in the proportion of converted tumor-migratory CD103⁺

**Figure 2. Radioimmunogenic tumors require conventional dendritic-cell migration for immunological responsiveness to radiation therapy.**
**(A)** Experiment design: mixed bone marrow (BM) chimeras were established for 8–10 wk, and then MC38 tumors were implanted into mice. Diphtheria toxin (DTx) was administered 3 d before radiation therapy (RT) and every 3 d after for a total of four doses to deplete conventional dendritic cells. **(B)** The TdLN was harvested 1 d after radiation and analyzed by flow cytometry. The number of (i) mig CD103$^+$ DC and (ii) mig CD11b$^+$ DC in the TdLN were quantified in WT::CCR7$^{-/-}$ (gray) or Zbtb46$^{dtr}$::CCR7$^{-/-}$ (white) BM chimeras treated with DTx. **(C)** Animal survival was monitored after treatment outlined in 2A for WT::CCR7$^{-/-}$ (gray) or Zbtb46$^{dtr}$::CCR7$^{-/-}$ (white) BM chimeras for NT (squares) or RT (circles) groups. **(D)** Individual tumor growth curves for animals from (C). Data represent the mean ± SEM of each group. Results shown are representative of two independent experiments with n = 5–8 animals/group. ***P < 0.001 and ****P < 0.0001.

cDC1s co-expressing CD40 and CD80 in the radiation-treated group compared with untreated controls (Figs S4 and 1D ii). The strong fluorescence of the Kaede transgene limited the number of maturations markers we could study on the DC in the TdLN; however, CD80 and CD40 are relevant and robust measures of DC maturation in lymph nodes (32). Thus, broadly, the DCs from radiation-treated animals were more mature than the tumor-migratory DC populations in the TdLN from untreated controls. Although there was some evidence that converted migratory CD11b$^+$ DCs from the radiation-treated group had an increased proportion of DCs co-expressing CD40 and CD80, this effect was significantly reduced in these CD11b$^+$ DCs compared with CD103$^+$ DCs (Figs S4 and 1E ii). These data suggest that although radiation therapy fails to increase the number of CD103$^+$ DCs migrating from the tumor, treatment does increase the proportion of tumor-migratory CD103$^+$ DCs co-expressing CD40 and CD80 in the TdLN relative to untreated controls. Importantly, these changes could not be identified in unconverted DC and were not detectable when assessing the total DC population of the TdLN (data not shown), emphasizing the importance of the Kaede model in studying these site-specific effects. In addition, although a range of inefficiencies mean that not all of the cells in the tumor are converted, by focusing only on converted cells, we ensure that we can be

confident that these cells originated in the UV and radiation treatment fields.

### CCR7-mediated cDC migration is required for the immunological efficacy of radiation therapy in radioimmunogenic tumors

Next, we set out to determine whether cDC migration was required for the efficacy of radiation therapy. The chemokine receptor CCR7 is up-regulated upon DC maturation, and this receptor has been shown to play an important role in guiding cDC1 migration from the tissue through the lymphatics to the dLN (33, 34). Animals that completely lack CCR7 have impaired T cell and DC migration, resulting in disrupted architecture within the LN (34). To overcome this issue and formally examine the role of CCR7-mediated migration, we developed a mixed bone marrow chimera approach (12). Animals were given a mixture of 50% CCR7$^{-/-}$ bone marrow and 50% bone marrow from mice where the human diphtheria toxin receptor (dtr) expression is driven by the cDC-specific transcription factor Zbtb46 (Zbtb46$^{dtr}$::CCR7$^{-/-}$, Fig 2A). Thus, the presence of the Zbtb46$^{dtr}$ bone marrow ensured that cDCs that can express CCR7 are present during animal and tumor development to ensure normal immune biology. However, we can deplete these CCR7$^{+/+}$ cDCs by administering diphtheria toxin, leaving behind only the CCR7$^{-/-}$

deficient cDCs at the time of radiation administration. This design ensured normal tumor and LN biology before and during experiments. As controls, we gave a second group of mice 50% WT (C57BL/6) bone marrow and 50% CCR7$^{-/-}$ bone marrow (WT::CCR7$^{-/-}$ Fig 2A). Bone marrow chimeras were allowed to reconstitute for 8–10 wk, and then tumors were implanted (Fig 2A). When tumors reached ~5 mm average diameter, animals were given diphtheria toxin to deplete cDCs expressing CCR7, leaving behind only CCR7$^{-/-}$ cDCs. Tumors were subsequently treated with 12 Gy of CT-guided radiation (Fig 2A). We harvested the TdLN 1 d after treatment for analysis by flow cytometry to confirm that DC migration was impaired. As expected, the number of migratory CD103$^+$ cDC1s in the TdLN were significantly reduced in the animals where cDC migration was impaired (Fig 2B i). We also saw a reduction in the number of migratory CD11b$^+$ cDC2s in the TdLN after treatment, although not to same degree as migratory CD103$^+$ cDC1s (Fig 2B ii). These data confirm that our mixed bone marrow chimera approach impairs migratory cDC migration to the TdLN. Next, we monitored tumor growth and animal survival when cDC migration was impaired during treatment with radiation (Fig 2C). When control animals (WT::CCR7$^{-/-}$) were treated with radiation, there was a significant survival advantage in these animals, and half of the tumors were cured as compared with untreated control animals (Fig 2C and D i). However, in animals where cDC migration was impaired (Zbtb46$^{dtr}$::CCR7$^{-/-}$), this survival advantage disappeared after radiation therapy, and tumor cures were no longer observed (Fig 2C and D ii). These data demonstrate that the efficacy of radiation therapy relies on CCR7-mediated cDC migration either at the time of or after radiation in radioimmunogenic MC38 tumors.

## Radiation impairs cDCs migration to the TdLN in poorly radioimmunogenic tumors

Our previous work had demonstrated that radiation failed to induce cDC maturation in poorly radioimmunogenic Panc02-SIY tumors (17). To follow DC migration in this tumor model, Panc02-SIY cells were implanted into Kaede mice, and when they reached 5–6 mm average diameter, tumors were photoconverted then treated with 12-Gy radiation (Fig 3A). In the Panc02-SIY tumor model, we again observed that the migratory CD103$^+$ and CD11b$^+$ DC populations contained the highest frequency of converted tumor-migratory DCs (Fig 3B). Interestingly, when we compared untreated and radiation-treated animals, we observed that radiation decreased the proportion of converted cells within migratory CD103$^+$ and CD11b$^+$ DC populations (Fig 3B). Next, we evaluated the number of converted migratory CD103$^+$ and CD11b$^+$ DCs in the TdLN and found significantly fewer converted CD103$^+$ and CD11b$^+$ DCs in the TdLN after treatment with radiation (Fig 3C i and D i). These data suggested that unlike the radioimmunogenic MC38 model, radiation impairs the migration of cDCs from the tumor to the TdLN in the poorly radioimmunogenic Panc02-SIY tumor model. Finally, we analyzed the expression of the co-stimulatory molecules CD40 and CD80 which are up-regulated upon DC maturation. In the radiation-treated group, we did not detect any differences in the expression of these markers in converted tumor-migratory CD103$^+$ DCs between untreated and radiation-treated groups (Fig 3C ii). We did note a slight increase in proportion of cells co-expressing CD40 and CD80 in the converted tumor-migratory CD11b$^+$ DCs, although the overall

number of these cells migrating was still significantly reduced in radiation-treated animals compared with untreated controls (Fig 3D). Notably, we did observe that in the radioimmunogenic MC38 tumor model that the converted (tumor derived) DCs are more mature at baseline (Fig 1D ii and C ii) than the converted DCs at baseline in the poorly radioimmunogenic Panc02-SIY model (Fig 3C ii and D ii). To ensure that these were not model-specific events, we assessed DC migration after radiation therapy in two additional head and neck tumor models, radioimmunogenic Moc1 and poorly radioimmunogenic Moc2 tumors (Fig S5A). Although the number of converted tumor-migratory CD103$^+$ DCs and CD11b$^+$ did not change in either group, there was an increased proportion of both DC subsets co-expressing CD40 and CD80 in the radiation-treated Moc1 group (Fig S5B i and ii and C i and ii). Although Moc2 is a poorly radioimmunogenic tumor (16, 17), Moc2 tumors exhibited an increase in CD103$^+$ DC but not CD11b$^+$ DC co-expressing CD40 and CD80 in the TdLN after radiation therapy (Fig S5B i and ii and C i and ii). Given that this tumor has an extremely poor infiltrate of T cells in the tumor (35), it is possible that the mechanisms driving poor radioimmunogenicity may not be related to DC maturation in this tumor model (16, 36). Thus, in the well-infiltrated Panc02-SIY model, treatment with radiation reduces the number of tumor-migratory cDCs that are found in the TdLN, and this may explain why the Panc02-SIY tumor model is poorly responsive to radiation therapy.

## Radiation increases the expression of genes associated with innate adjuvant signaling in cDCs from radioimmunogenic tumors

We have previously shown that radiation induced cDC maturation within the tumor in radioimmunogenic MC38 tumors, and this process failed to occur in poorly radioimmunogenic Panc02-SIY tumors (17). Moreover, data from the Kaede photoconvertible mice demonstrated that cDC migration is impaired in poorly radioimmunogenic tumors where cDCs failed to mature. Thus, we set out to identify which pathways might be differentially regulated between cDCs from radioimmunogenic MC38 tumors and poorly radioimmunogenic Panc02-SIY tumors after radiation therapy using a transcriptional-based approach. MC38 or Panc02-SIY tumors were established in mice, treated with radiation, and CD45$^+$ cells from the tumor were analyzed using the 3′ Gene Expression single-cell RNA sequencing platform from 10X Genomics (Fig 4A). To identify cDCs, we used an unsupervised clustering algorithm from the Loupe Cell Browser (10X Genomics). Using the graph-based cluster algorithm, 18 clusters were identified, with cluster 16 expressing many cDC-related genes such as Flt3, Itgae, Tlr3, and Zbtb46 (Fig 4B and C). These cDC populations were distinct from markers denoting other key immune populations across the tumor-infiltrating cells (Fig S6). To define cDCs for our analysis, we used cells that were in cluster 16 and expressed one or more Zbtb46 transcripts (Figs 4D and S7A). Recent studies have identified multiple DC populations in human and murine tumors with distinct transcriptional profiles (32, 37). To identify these profiles in our tumors, we again used the clustering algorithm from the Loupe Cell Browser to separate the DC into distinct populations (Fig S8). We then analyzed the expression of the distinctive genes that separate DC subpopulations in these studies. Using this approach, we can identify that cluster 3 has more features of mregDC, with expression of the activation markers Ccr7,

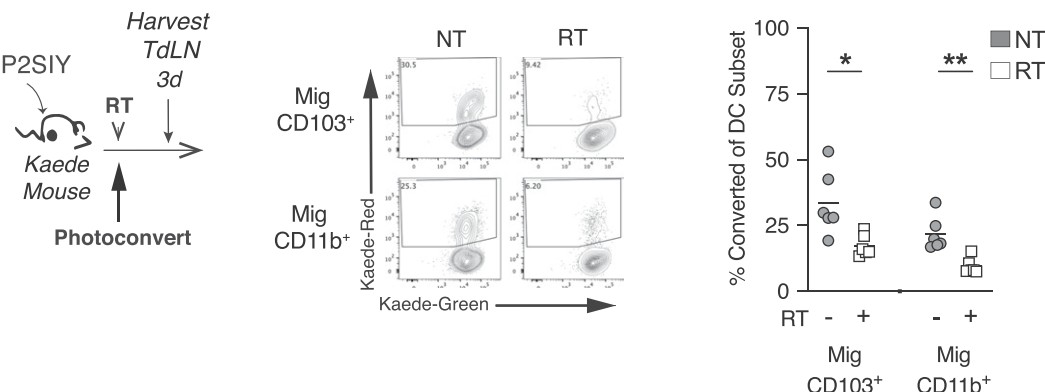

*A) Experiment Design*   *B) Converted Migratory DCs (Kaede-Red)*

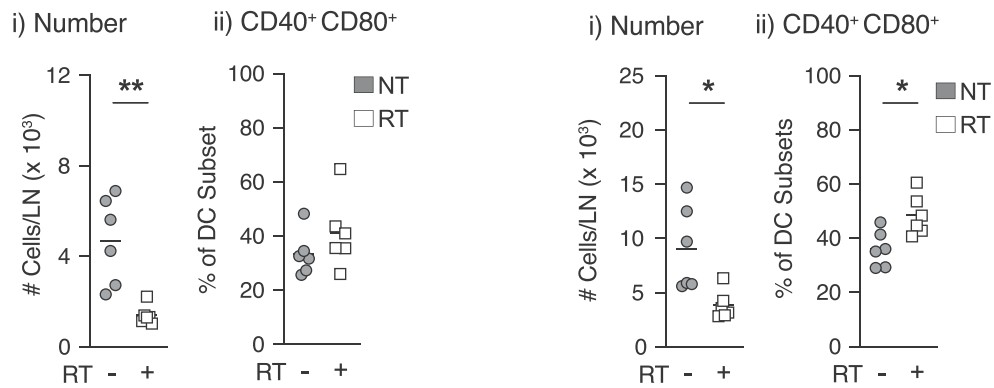

*C) Converted (Kaede-Red) CD103⁺ cDC1s*   *D) Converted (Kaede-Red) CD11b⁺ cDC2s*

**Figure 3.   In poorly radioimmunogenic tumors, radiation therapy impairs conventional dendritic cell migration from the tumor to the TdLN.**
**(A)** Experiment design: Panc02-SIY (P2SIY) tumors were established in Kaede mice. When tumors reached an average of 5–6 mm average diameter, they were photoconverted and then treated ±12-Gy CT-guided radiation therapy, and the TdLN was harvested for analysis by flow cytometry at indicated timepoints. **(B)** Representative flow plot from 3 d post-photoconversion for migratory (mig) CD103⁺ and CD11b⁺ DC populations in the TdLN showing converted (Kaede-red⁺) DCs. For untreated (NT, gray) or radiation (RT, white) treatment, the proportion converted Kaede-red⁺ DCs within each DC subset in the TdLN: (i) Mig CD103⁺ and (ii) Mig CD11b⁺. **(C)** (i) The absolute number of Kaede-red⁺ CD103⁺ DC/dLN and (ii) frequency of CD40⁺ CD80⁺ within converted CD103⁺ DCs for NT (gray) or RT (white) groups. **(D)** (i) The absolute number of Kaede-red⁺ CD11b⁺ DC/dLN and (ii) frequency of CD40⁺ CD80⁺ within converted CD103⁺ DCs for NT (gray) or RT (white) groups. Data represent the mean ± SEM of each group. Results shown are representative of two independent experiments with n = 4–6 animals/group. *P < 0.05 and **P < 0.01.

Fscn1, and Relb, whereas cluster 2 has more features of DC1, with expression of Naaa, Cadm1, and Xcr1 (Fig S8) (32). By contrast, cluster 1 has features of DC2, with higher expression of Sirpa, H2-DMb2, and Itgam. These data indicate that our cDC population is consistent with other defined DC populations by single-cell RNA-Seq. As an initial analysis of the DC in our tumors, we compared differences in gene expression that occur between the DC-infiltrating different tumors at baseline and after radiation therapy (Fig S7A). Of the 15 genes that were up-regulated after radiation therapy in both tumors, these were mapped to the cellular response to radiation (Cdkn1a), interferon γ (Cd274), and components of the S100 family (Fig S7C). To focus on the differences in gene expression that occur after radiation in cDCs for each tumor type (Fig S7B) using differential gene expression analysis, we identified that in MC38 cDCs, there were 93 genes that were up-regulated with radiation and 116 genes in Panc02-SIY cDCs (Fig S7B). This analysis revealed that the cyclin-dependent kinase inhibitor gene Cdkn1a

was significantly increased in both MC38 and Panc02-SIY cDCs after radiation therapy (Fig S7C). Cdkn1a functions as part of the DNA damage response pathway and is known to be up-regulated after radiation (38, 39). Thus, these data indicate that the cDC included in our analysis were likely in the tumor at the time of treatment with radiation therapy.

To understand the maturation differences after RT, we compared MC38 versus Panc02-SIY radiation-treated cDCs. We identified 370 differentially expressed genes (Figs 4E and S7). To obtain a global overview of the connections between the genes up-regulated in MC38 cDCs, we analyzed the data using Ingenuity Pathway Analysis (IPA) from QIAGEN. First, we looked at immune-related canonical pathways that were predicted to be altered in MC38 versus Panc02-SIY-treated cDCs. This analysis revealed that after radiation, MC38 cDCs express more genes associated with "dendritic cell maturation" and Panc02-SIY cDCs expressed more genes associated with the "PD-1, PD-L1 cancer immunotherapy pathway" (Fig 4F). We then

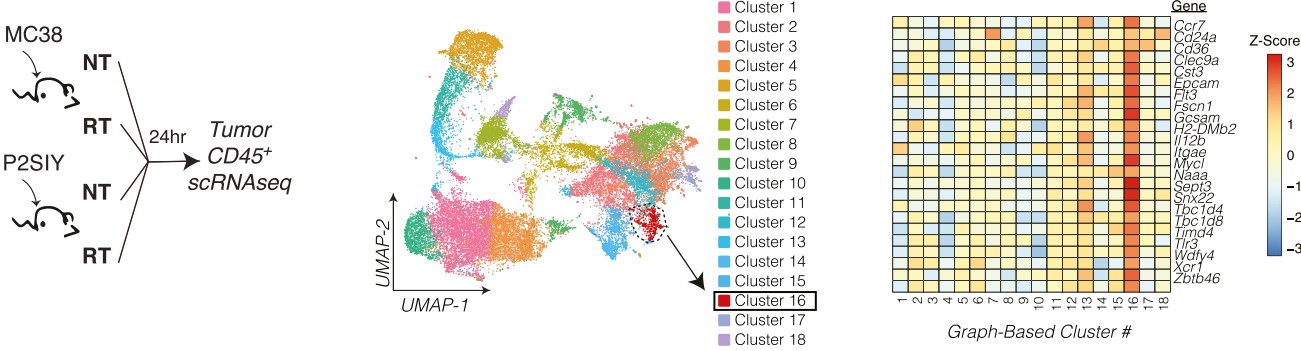

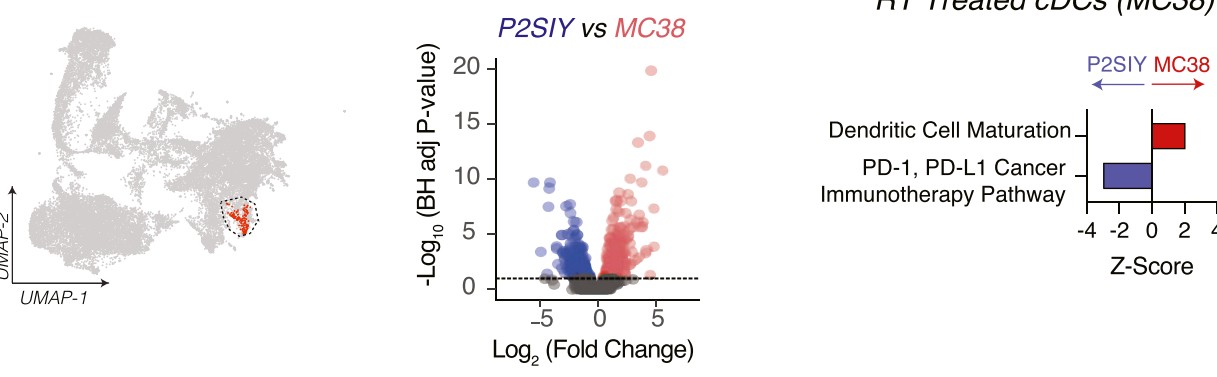

## G) IPA Upstream Analysis RT Treated cDCs (MC38)

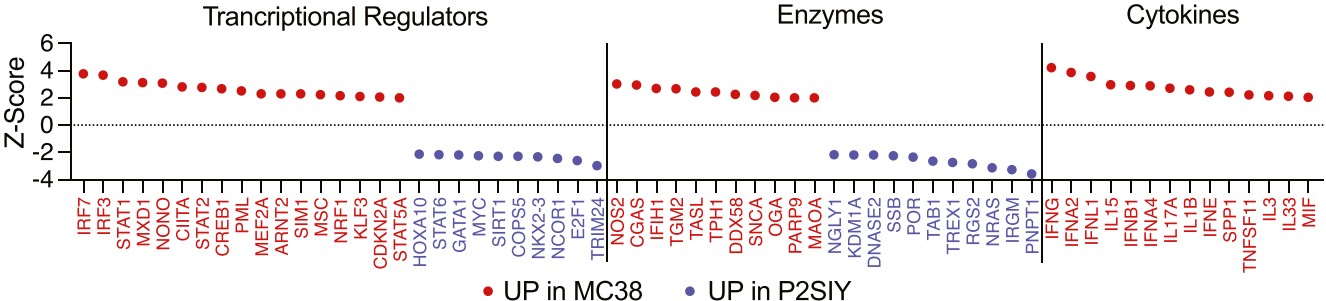

**Figure 4. Increased expression of genes associated with innate adjuvant signaling in conventional dendritic cells (cDCs) after radiation in radioimmunogenic tumors.**
**(A)** Experiment design: MC38 or Panc02-SIY (P2SIY) tumors were implanted into WT mice and then treated with ±12 Gy of radiation therapy when they reached 5–6 mm average diameter. Twenty-four hours posttreatment, tumors were harvested, and CD45+ were processed for single RNA sequencing (scRNAseq). n = 3 animal/group. **(B)** Graph-based clustering in the Loupe Cell Browser from 10X Genomics (UMAP plot). Each dot represents an individual cell. **(C)** Heatmap showing the z-score of expression for cDC-specific genes across each cluster that was identified in (B). **(D)** Cells in cluster 16 that expressed one or more transcripts of Zbtb46 were labeled in red and used for downstream analysis and comparisons. Each dot represents an individual cell. **(E)** cDCs from radiation-treated MC38 tumors were compared with cDCs from P2SIY tumors using differential expression, and the data were plotted on using a volcano plot. Each dot represents an individual gene, with the color indicating genes that are significantly increased in MC38 cDCs (red), significantly increased in P2SIY cDCs (blue) and not significant (gray). Genes were considered significant with BH-adjusted P-value < 0.1 and fold change > 1.3. **(F)** The significant genes (red + blue dots) were input in the Ingenuity Pathway Analysis (IPA) program from QIAGEN to identify canonical pathways related to "cellular immune response." **(G)** Within IPA, upstream analysis was performed to identify transcriptional regulator, enzymes, and cytokines that are predicted to be activated or inhibited in corresponding cDC populations. Dots represent significant upstream regulators and their corresponding Z-score. Red dots predict activation in radiation-treated MC38 cDCs, and blue dots predict activation in P2SIY tumors. Pathways and regulators were considered significant with P-value > 0.05 and absolute z-score > 2.0 in IPA.

used the upstream regulator analysis for immune-based pathways in IPA to identify transcriptional pathways, enzymes, and cytokines that are predicted to be altered in cDCs after treatment with radiation (Fig 4G). Within cDCs from MC38 tumors, the transcription factors IRF3, IRF7, and STAT1 were predicted to be active given the pattern of increased gene expression in these cells (Fig 4G). IRF3 and IRF7 are known to be activated during innate immune responses and signal downstream of many pattern recognition receptors within immune cells (40, 41). This analysis also suggested that enzyme-important nucleic acid sensing in cells (CGAS, DDX58) were increased in MC38 cDCs (Fig 4). Further supporting the hypothesis that MC38 cDCs receive maturation signals after radiation, IPA predicted activation by type I interferons (IFNA2, IFNA4, IFNB1), type II interferons (IFNG), and IL1B in MC38 cDCs relative to Panc02-SIY cDCs (Fig 4F). These data suggest that innate adjuvant signaling pathways are likely activated in cDCs from MC38 tumors treated with radiation therapy, and this leads to successful DC maturation. To determine whether the candidate upstream regulators were altered in the tumor environment, we analyzed all scRNASeq transcripts for significant changes between irradiated Panc02-SIY versus irradiated MC38 tumors (Table S1 and Supplemental Data 1). Within these genes, we highlighted those identified as candidate upstream mediators by IPA (Fig S9). These genes show a mixed pattern. Some candidate genes such as SPP1, IRF7, and IL1B are expressed more in MC38 tumors and may be candidate mediators of altered DC maturation. By contrast, others such as IFNG and IL17A are expressed less in MC38 tumors, suggesting that they are not candidate mediators of altered DC maturation that is predicted by IPA analysis. Further studies are necessary to identify the mechanisms resulting in altered DC maturation after radiation therapy.

### Exogenous adjuvant increases intratumoral cDC migration to the TdLN in poorly radioimmunogenic tumors

The comparative transcriptional analysis of cDCs after radiation indicated that DCs in Panc02-SIY tumors either fail to receive maturation signals or these signals are suppressed after radiation. We have previously shown that in poorly radioimmunogenic tumors, the combination of the exogenous adjuvant poly I:C with radiation therapy leads to enhanced intratumoral cDC maturation and improved responses to treatment when compared with radiation alone (17). To understand whether the innate adjuvant poly I:C induced intratumoral DC migration in this model, we used the Kaede photoconvertible mice to follow DC trafficking to the TdLN. Poorly radioimmunogenic Panc02-SIY tumors were established in Kaede mice and then photoconverted from Kaede-green to Kaede-red (Fig 5A). This was followed by treatment with radiation and intratumoral poly I:C (Fig 5A). The TdLN was harvested 1 d after treatment to assess tumor-migratory DC populations by flow cytometry (Fig 5A). We observed that treatment with poly I:C significantly increased the proportion of CD103$^+$ cDC1 within converted (Kaede-red) DCs, whereas decreasing the proportion of converted CD11b$^+$ cDC2 in the TdLN (Fig 5B). When the absolute numbers of tumor-migratory DCs were analyzed, we observed significant increases in the total number of Kaede-red CD103$^+$ cDC1 and CD11b$^+$ cDC2s (Fig 5C i and D i). Similarly, treatment with poly I:C significantly increased the total number of Kaede-red CD103$^+$ cDC1 and CD11b$^+$

cDC2s in the TdLN of MC38 tumors (Fig S10). These data indicate that exogenous administration of poly I:C is capable of driving cDC migration from the tumor to TdLN in both radioimmunogenic and in poorly radioimmunogenic tumors. Furthermore, when we analyzed the expression of the CD80 on tumor-migratory (Kaede-red) DCs, we noted a significant increase in the frequency and intensity of this maturation marker on both CD103$^+$ and CD11b$^+$ cDCs (Fig 5C ii and iii and D ii and iii). In each case, poly I:C affected DCs independent of radiation effects. Taken together, these data demonstrate that exogenous poly I:C overcomes the impaired intratumoral DC migration that occurs in poorly radioimmunogenic tumors treated with radiation. These data support the hypothesis that the innate adjuvant balance in tumors is critical to support adaptive immune responses after radiation therapy (42) but also demonstrate that cancer cells can establish distinct environments in genetically identical mice that can suppress innate and therefore adaptive immunity to tumors. To understand the effects of the tumor environment on DC that migrated from the tumor versus the effects that both converted and unconverted DC may experience in the TDLN, we analyzed the maturation status of converted versus unconverted migratory CD103$^+$ DC and CD11b$^+$ DCs in the TdLN of MC38 tumors treated with radiation therapy and pIC. In untreated tumors, CD80 is not significantly different between converted and unconverted DCs (Fig S11). However, after radiation and pIC treatment, there is a dramatic increase in CD80 that is almost entirely confined to the converted DC populations. These data suggest that the effect of tumor treatment dominantly occurs in the converted population and is unlikely to be occurring in the TdLN where unconverted DC would receive an equal maturation signal.

## Discussion

A long-standing question within the field of radiation oncology is whether radiation therapy can function as an in situ vaccine capable of priming new CD8$^+$ T-cell responses directed against tumors. To function in this capacity, treatment with radiation therapy would likely require mature cDC1s to carry tumor-associated antigens from the tumor to the TdLN for cross-presentation to T cells, and to date, there has been limited direct evidence supporting the role of DC migration in radiation. Here, we demonstrate that cDC migration from the tumor to the TdLN is necessary for the immunological efficacy of radiation in the radioimmunogenic MC38 tumor models. This is one of the first steps in generating tumor-specific immunity, and these data suggest that radiation has the capacity to initiate the process of priming new CD8$^+$ T-cell responses. Once in the TdLN, these tumor-migratory cDCs may either function to directly cross-present antigens to antigen-specific CD8$^+$ T cells, or they may instead hand off antigen to other cDC1 subsets in the LN for cross-presentation (12, 43). Our previous work indicated that radiation can drive intratumoral cDC1 maturation in radioimmunogenic tumors (17). These new studies build on this work and use clear genetic models to provide convincing evidence that radiation is capable of inducing both cDC maturation and migration to the TdLN after treatment and that these events are critical to maximizing the benefits or radiation therapy.

## A) Experiment Design

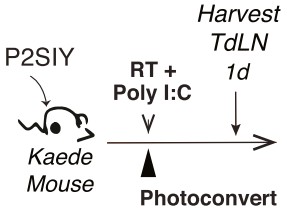

## B) Frequency of Converted (Kaede-Red)

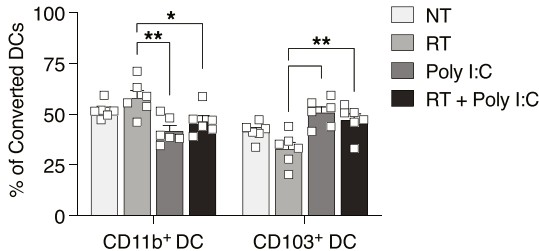

## C) Converted (Kaede-Red) CD103⁺ DCs

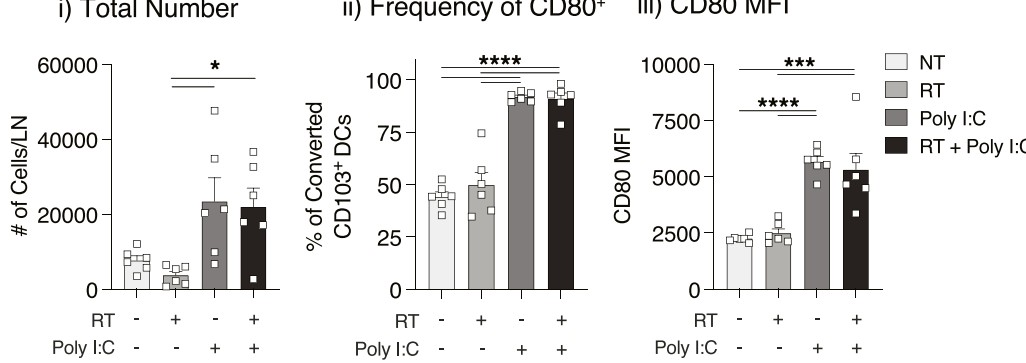

## D) Converted (Kaede-Red) CD11b⁺ DCs

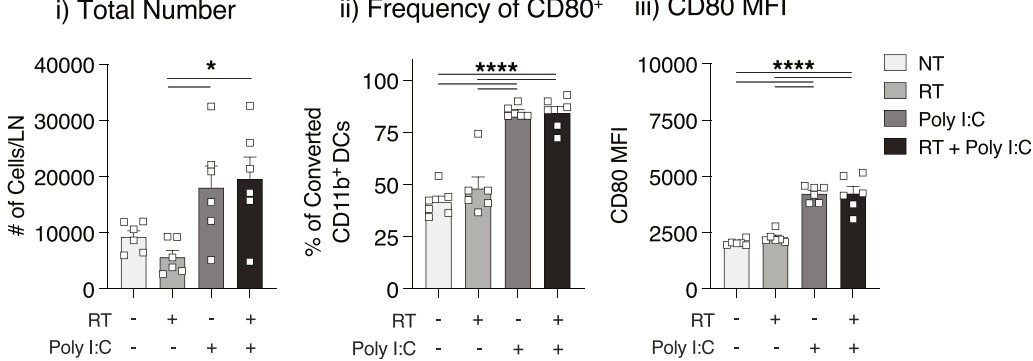

**Figure 5. In poorly radioimmunogenic tumors, exogenous adjuvant increases the number of tumor-migratory conventional dendritic cells in the TdLN.**
**(A)** Experiment design: Panc02-SIY (P2SIY) tumors were established in Kaede mice. When tumors reached 5–6 mm average diameter, they were photoconverted with violet light and treated with 12-Gy radiation in combination 50 μg intratumoral poly I:C. **(B)** Gating on converted (Kaede-red⁺) DCs (CD11c⁺ MHC-II⁺) and the frequency of mig CD103⁺ versus mig CD11b⁺ conventional dendritic cells were assessed. **(C)** The TdLN was harvested 1 d after treatment to assess (i) the total number of converted mig CD103⁺ DCs, (ii) frequency of converted mig CD103⁺ DCs expressing CD80, and (iii) the median fluorescence intensity (MFI) on CD103⁺ DCs. **(D)** The TdLN was harvested 1 d after treatment to assess (i) the total number of converted mig CD11b⁺ DCs, (ii) frequency of converted mig CD11b⁺ DCs expressing CD80, and (iii) the median fluorescence intensity (MFI) on CD11b⁺ DCs. Data represent the mean ± SEM of each group. Results shown are representative of two independent experiments with n = 6 animals/group. *P < 0.05, ***P < 0.001 and ****P < 0.0001.

In our tumor models, the inguinal LN drains the tumor in addition to other surrounding tissues which makes it difficult to determine whether migratory DC populations in the TdLN came from the tumor or other surrounding tissues (27, 28). By using the Kaede photoconvertible mice, we were able to specifically identify tumor-migratory DC populations within the TdLN that came from the treatment field. Similar to previous reports, DCs were one of the main cell types migrating from the tumor to the TdLN, and converted cells were predominantly found in populations that we identified as being migratory DCs using flow cytometry markers (27, 28). These data confirm that our flow cytometry panel accurately identifies migratory DC populations. These data also highlight why it has been difficult to determine how different treatments impact DC migration from the tumor because the migratory cells originating in the tumor make up a relatively small proportion of total migratory DC subsets in the TdLN. Thus, the Kaede mice are a powerful tool to

specifically track and characterize changes in DC populations migrating from the tumor to the TdLN.

Critically, our data also suggest that radiation has differential impacts on DC migration, depending on the tumor model being studied. We found that radiation does not change the number of tumor-migratory DCs in the TdLN in the radioimmunogenic MC38 tumor model. By contrast, treatment with radiation impaired DC migration to the TdLN in poorly radioimmunogenic Panc02-SIY tumors. It remains possible that the kinetics of migration in the poorly radioimmunogenic model is different than in radioimmunogenic tumors. However, in radioimmunogenic tumors, radiation had no impact on the migration across several timepoints, indicating that the impairment in DC migration might be a sustained phenomenon in the Panc02-SIY tumor model. CD40 and CD80 are widely used to define mature DC as they are consistently up-regulated adjuvant signals (31, 44, 45). Although it would be expected the most of DCs found in the LN have a "mature" phenotype because they were able to successfully migrate from the tissue, it is likely that migrating DCs exist on a spectrum of maturity. This maturity level likely determines their ability to successfully cross-present antigens and to activate adaptive immune responses. For instance, it has been reported that semi-mature DCs are capable of migrating from the tissue to the dLN; however, these DCs are poor stimulators of T-cell immunity (33). These data may explain why modest increases in DC maturation after radiation can have significant impacts on outcome. Future studies are needed to perform a comprehensive analysis to identify additional migratory DCs maturation markers so that we can determine whether more subtle difference exist between the migratory DCs. Our scRNASeq analyses show that DC differences in DC maturation can be detected very early after radiation therapy. Notably, we could not see significant changes in type I IFN genes after radiation therapy. Although it remains possible that we do not see these changes because we only profile the CD45$^+$ infiltrating immune cells, MC38 tumors have been shown to up-regulate type I IFN 3 d after radiation therapy, and this is dependent on STING expression in host cells (46), suggesting a host origin for type I IFN transcripts. In this same manuscript, the authors demonstrated that macrophages and DCs were the major source of type I IFN transcripts (46). STING activation in host cells has been shown to be dependent on the timeline of micronuclei formation in the cancer cells (47), which can take 3–7 d to generate STING activation in vitro. Given the absence of detectable type I IFN changes in our samples at d1 post-RT and the expected slow timeline of adjuvant activation in the tumor environment, we do not currently have a good explanation for this early maturation effect or RT on DC in MC38 tumors as profiled in the scRNASeq. Importantly, it is unclear whether there is active suppression of DC maturation in poorly radioimmunogenic tumors, an absence of positive maturation signals, or some combination of both. There are many potential mechanisms that could be responsible for the suppression of DC maturation after treatment, whether coming from directly from the tumor or indirectly through differential intratumoral immune populations, for example, macrophages (48, 49, 50, 51). These regulatory features may include metabolic features of the tumor—for instance, it has been reported that oxidized cholesterol ligands secreted from tumors can suppress the expression of CCR7 in DCs-, and this impairs the migration

from the tumor to TdLN (52). However, the exogenous delivery of maturation stimuli by delivery of adjuvants such as pIC (17) or STING ligands (53) can improve T-cell control of tumors after radiation therapy, suggesting that if active suppression of DC maturation is in place, it can be overcome. Taken together, these data suggest that individual tumor microenvironments determine whether DCs are activated or suppressed after treatment with radiation, and this in turn determines whether DCs migrate to the TdLN to cross-present antigens. Additional work is needed unravel the tumor-specific mechanisms that prevent DCs from maturing and in this manner, prevent the priming of tumor-reactive CD8$^+$ T cells in the TdLN.

These data are focused on DC migration from implanted subcutaneous tumors. However, DC migration from spontaneously emerging tumors may be differently regulated (54, 55). Moreover, the direct injection of cancer cells into mice can result in a strong immune response that can impact the subsequent immune biology of the tumor (6, 16), which may be absent in an induced tumor (55). For these reasons, it can be critical to explore the dynamics of immunity over time in preclinical models to understand whether they are directly applicable to patient tumors (19). In addition, these data suggest that we refocus our attention on the T cells in the TdLN that recognize the antigens cross-presented by DC as an alternative to focusing on the exhausted T-cell populations that are uniquely found in the tumor environment. These T cells in the different locations may have different phenotypes and may have different co-stimulation requirements that could be uniquely targeted to improve their expansion and effector function after radiation. Recent data suggests that abscopal responses may be more dependent on a stem-like population emerging from the lymph node, rather than tumor resident cells (56). Related to these data, the therapeutic targets may be distinct depending on whether the goal of local treatment is local or distant. The role of the tumor draining lymph node and circulating T-cell populations has been shown to be distinct depending on whether the therapy includes checkpoint inhibitors (6, 23) and whether responses are measured at the treatment site or distant disease (56, 57). These data suggest that local therapy amplified by checkpoint inhibitors may occur without the contribution of lymph node responses (6, 22, 23), but to generate systemic antitumor immunity amplified by radiation therapy, lymph node responses are important (56).

Critically, radiation is an essential partner in these processes. In our studies, innate adjuvants could dramatically activate DCs but were unable to impact tumor growth. Radiation provides the essential transfer of cell-associated antigens to phagocytic cells, and the milieu of endogenous adjuvants and counterregulatory cells and pathways is the tumor-specific feature that decides the final immune impact of radiation therapy.

## Materials and Methods

### Animals and cell lines

Experiments used 6–8 wk old C57BL/6 (#000664), B6.SJL (#002014), CCR7$^{-/-}$ (#006621), and Zbtb46$^{dtr}$ (#019506) mice that were obtained from the Jackson Laboratories. Kaede transgenic mice were kindly provided by Dr. Amanda Lund at Oregon Health and Science

University (26). Kaede transgenic mice were crossed with CCR7$^{-/-}$ mice in house to generate Kaede CCR7$^{-/-}$ animals for bone marrow chimera experiments. 2C TCR transgenic mice were kindly provided by Dr. Thomas Gajewski at the University of Chicago. Survival experiments were performed with 8–14 mice per experimental group and mechanistic experiments with 4–6 mice per group. Animal protocols were approved by the Earle A Chiles Research Institute (EACRI) Institutional Animal Care and Use Committee (Animal Welfare Assurance No. D16-00526). The Panc02-SIY pancreatic adenocarcinoma line expressing the model antigen SIY was kindly provided by Dr. Ralph Weichselbaum at the University of Chicago. MC38 colorectal carcinoma line was obtained from Dr. Kristina Young at EACRI. Moc1 and Moc2 oral squamous-cell carcinoma lines were kindly provided by Dr. Ravindra Uppaluri at the Dana Faber Cancer Institute. Panc02-SIY, Moc1, and Moc2 cell lines were grown in complete RPMI containing 10% heat-inactivated FBS, 100 U/ml penicillin, and 100 $\mu$g/ml streptomycin. MC38 cell lines were grown in DMEM containing 10% heat-inactivated FBS, 100 U/ml penicillin, and 100 $\mu$g/ml streptomycin. Pathogen and mycoplasma contamination testing was performed on all cell lines within the past 6 mo using the IMPACT II Mouse PCR Profiling from IDEXX BioAnalytics.

### Tumor treatments

Tumors were implanted subcutaneously into the right flank as follows: $2 \times 10^5$ MC38, $5 \times 10^6$ Panc02-SIY, $1 \times 10^6$ Moc1, and $1 \times 10^5$ Moc2. When tumors were ~5 mm in average diameter, mice were randomized to receive treatment with CT-guided radiation using the Small Animal Radiation Research Platform (SARRP) from Xstrahl. Dosimetry was performed using Murislice software from Xstrahl. The SARRP delivered a single dose of 12 Gy to an isocenter within the tumor using a $10 \times 10$ mm collimator and a 45° beam angle to minimize dose delivery to normal tissues. For photoconversion experiments using the Kaede mice, tumors were converted as described by Steele et al (28). Briefly, tumors were implanted in animals with shaved skin and for photoconversion animals, were completely covered in aluminum foil except for the tumors which were exposed to 405-nm LED light source using a collimator for 5 min (Prizmatix). Where radiation therapy followed Kaede conversion, the UV conversion in Kaede mice was performed immediately before radiation therapy, with at most a 1 h lapse between conversion and radiation treatment. In all survival experiments, tumor length and width were measured 2–3 times per week using calipers. Mice were euthanized when tumor size exceeded 12 mm in any dimension or when body condition score declined 1 level.

### Tissue processing

After dissection, tumors were weighed and minced into small fragments, then transferred into C tubes from Miltenyi Biotec containing enzyme digest mix with 250 U/ml collagenase IV (#LS004188; Worthington Biochemical), 30 U/ml DNase I (#4536282001; Millipore-Sigma), 5 mM CaCl$_2$, 5% heat-inactivated FBS, and HBSS. Tissue was dissociated using a gentleMACS tissue dissociator from Miltenyi Biotech. This was followed by incubation at 37°C for 30 min with agitation. For the dLNs, capsules were cut open and incubated with enzymatic mix described above at 37°C for

15 min with agitation. Enzyme mix containing dLNs was then vigorously pipet mixed and incubated at 37°C for an additional 15 min. Enzymatic reactions for both the tumor and dLN were quenched using ice-cold RPMI containing 10% FBS and 2 mM EDTA. Single-cell suspensions were then filtered through 100-$\mu$m (tumor) or 40-$\mu$m (dLN) nylon cell strainers to remove macroscopic debris. Cells were washed and counted as described above.

### Flow cytometry

For staining, $2 \times 10^6$ cells were stained with Zombie Aqua Viability Dye from BioLegend (#423102) in PBS for 10 min on ice, and then Fc receptors were blocked with $\alpha$-CD16/CD32 antibodies from BD Biosciences (2.4G2) for an additional 10 min. After centrifugation, the supernatant was removed and cell were stained with a surface antibody cocktail containing in FACS buffer (PBS, 2 mM EDTA, 2% FBS) and Brilliant Stain Buffer Plus from BD Biosciences (#566385) for 20 min on ice. The following antibodies were purchased from BioLegend: F4/80-PerCP/Cy5.5 (BM8), CD11c-PE/Cy7 (N418), CCR7-PE (4B12), CD90.2-A700 (30-H12), CD19-A700 (6D5), MHC-II-BV421 (M5/114.14.2), CD11b-BV605 (M1/70), CD8$\alpha$-BV650 (53-6.7), and Ly6C-BV711 (HK1.4). CD40-FITC (HM40-3), CD103-APC (2E9), and CD24-APC e780 (M1/69) were obtained from Thermo Fisher Scientific. CD80-PE CF594 (16-10A1) and CD45-BV786 (30-F11) were purchased from BD Biosciences. After surface staining, cells were washed in FACS buffer and fixed for 20 min on ice with Fixation/Permeabilization Buffer from BD Biosciences (#554722). All samples were resuspended in FACS buffer and acquired on a BD Fortessa flow cytometer. Data were analyzed using FlowJo software from Tree Star, v10.7. cDC1 in the tumor were gated as leukocytes/single cells/live/CD45$^+$/CD90.2$^-$CD19$^-$/Ly-6C$^-$/MHC-II$^+$/CD24$^+$F4-80$^-$/CD11b$^-$/CD103$^+$. In the TdLN, migratory CD103$^+$ cDC1 were gated as leukocytes/single cells/live/CD45$^+$/CD90.2$^-$CD19$^-$/Ly-6C$^-$/MHC-II$^+$ CD11c$^+$/CD8$\alpha^-$/CD103$^+$, and resident CD8$\alpha^+$ cDC1 were gated as leukocytes/single cells/live/CD45$^+$/CD90.2$^-$CD19$^-$/Ly-6C$^-$/MHC-II$^+$ CD11c$^+$/CD103$^-$/CD8$\alpha^+$. cDC2 in the tumor were gated as leukocytes/single cells/live/CD45$^+$/CD90.2$^-$CD19$^-$/Ly-6C$^-$/MHC-II$^+$/CD24$^+$F4-80$^-$/CD103$^-$/CD11b$^+$. In the TdLN, migratory CD11b$^+$ cDC2 were gated as leukocytes/single cells/live/CD45$^+$/CD90.2$^-$CD19$^-$/Ly-6C$^-$/MHC-II$^{high}$ CD11c$^+$/CD8$\alpha^-$/CD103$^-$/CD11b$^+$, and resident CD11b$^+$ cDC2 were gated as leukocytes/single cells/live/CD45$^+$/CD90.2$^-$CD19$^-$/Ly-6C$^-$/MHC-II$^{int}$ CD11c$^+$/CD8$\alpha^-$/CD103$^-$/CD11b$^+$.

### Bone marrow chimeras

Bone marrow chimeras were generated using B6.SJL recipient mice that were irradiated with 10 Gy of radiation. Bone marrow cells were isolated from WT C57BL/6, CCR7$^{-/-}$, Zbtb46$^{dtr}$, or Kaede CCR7$^{-/-}$ donor mice femurs and tibias using a 27-G needle. Cells were filtered through a 70-$\mu$m cell strainer to generate a single-cell suspension and resuspended in PBS. Recipient mice received 1.5–2.5 $\times$ 10$^6$ of each specified donor bone marrow cells for a total of 3–5 $\times$ 10$^6$ cells/recipient animal that were transferred by retro-orbital injection. Tumors were implanted 8–10 wk after bone marrow reconstitution. Diphtheria toxin from Millipore-Sigma (#D0564) was administered 3 d before radiation at 20 ng/g intraperitoneally for initial DC depletion. This was followed by an additional 3 doses of

5 ng/g of diphtheria toxin that were given every 3 d to maintain depletion.

## Single-cell RNA sequencing

Panc02-SIY or MC38 tumors were treated ±12 Gy radiation as described above. Tumors were harvested 24 h posttreatment (n = 3 animals/group), processed into a single-cell suspension as described above and magnetically enriched using $CD45^+$ TIL MicroBeads (Miltenyi Biotec). Enriched cells were labeled with viability dye and CD45-APC. Live $CD45^+$ cells were sorted using a 100-$\mu$M nozzle on a BD Biosciences Aria cell sorter, and cells were processed according to the manufacturers protocol for the Chromium Single Cell 3′ Reagent Kit (v3.0) from 10X Genomics. Libraries were sequenced using an Illumina NovaSeq 6000 using the NovaSeq 6000 S2 Reagent Kit (v1.0). Data were processed using the Cell Ranger pipeline (v3.1) and subsequently analyzed with the Loupe Browser from 10X Genomics (v5.0). Using the Loupe Browser, differentially expressed genes between groups were considered significant if fold change of gene expression was > 1.3, and the Benjamini–Hochberg adjusted $P$-value was < 0.1. IPA software was from QIAGEN (v01-19-00), using default settings for Core Analysis. IPA canonical pathways related to "cellular immune response" with an absolute z-score greater than 2.0 and –log($P$-value) greater than 1.3 were considered significant.

## Statistics

Data were analyzed and graphed using Prism from GraphPad Software (v9.0). Individual data sets were compared using $t$ test, and analysis across multiple groups was performed using one-way ANOVA with individual groups assessed using Tukey's comparison. Kaplan–Meier survival curves were compared using a log-rank test.

# Data Availability

The data discussed in this publication have been deposited in NCBI's Gene Expression Omnibus (58) and are accessible through GEO Series accession number GSE201026 (https://www.ncbi.nlm.nih.gov/geo/query/acc.cgi?acc=GSE201026).

# Supplementary Information

# Acknowledgments

The work was funded by research grants National Institutes of Health (NIH) R01CA182311 (MJ Gough), R01 CA244142 (MJ Gough), NIH R01CA208644 (MR Crittenden), and by Providence Foundations of Oregon. Funders had no role in the preparation or content of the manuscript.

## Author Contributions

TC Blair: conceptualization, data curation, formal analysis, investigation, visualization, methodology, and writing—original draft, review, and editing.
S Bambina: investigation.
GF Kramer: investigation.
AK Dowdell: formal analysis, investigation, and visualization.
AF Alice: investigation.
JR Baird: investigation.
AW Lund: resources, supervision, and methodology.
BD Piening: supervision, methodology, and writing—original draft.
MR Crittenden: conceptualization, supervision, funding acquisition, and writing—original draft, review, and editing.
MJ Gough: conceptualization, formal analysis, supervision, funding acquisition, visualization, project administration, and writing—original draft, review, and editing.

## Conflict of Interest Statement

The authors receive research funding from Bristol Myers Squibb and VIR Biotechnology and consulting fees from Roche that is not directly related to the topic of this manuscript. Funders had no role in the representation of the data or the preparation of the manuscript.

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
