## [Reviewer comments · Life Science Alliance]

Life Science Alliance

Fluorescent tracking identifies key migratory dendritic cells in the lymph node after radiotherapy

Tiffany Blair, Shelly Bambina, Gwen Kramer, Alexa Dowdell, Alejandro Alice, Jason Baird, Amanda Lund, Brian Piening, Marka Crittenden, and Michael Gough

DOI: <https://doi.org/10.26508/lsa.202101337>

Corresponding author(s): Michael Gough, Providence Portland Medical Center

Review Timeline:

Submission Date:	2021-12-14
Editorial Decision:	2022-01-21
Revision Received:	2022-03-15
Editorial Decision:	2022-04-07
Revision Received:	2022-04-19
Accepted:	2022-04-20

Scientific Editor: Novella Guidi

Transaction Report:

January 21, 2022

Re: Life Science Alliance manuscript #LSA-2021-01337-T

Dr. Michael J Gough
Providence Portland Medical Center
Earle A Chiles Research Institute
4805 NE Glisan St
Portland, OR 97213

Dear Dr. Gough,

Thank you for submitting your manuscript entitled "Fluorescent tracking identifies tumor migratory dendritic cells as key for cross-presentation in the lymph node after radiotherapy" to Life Science Alliance. The manuscript was assessed by expert reviewers, whose comments are appended to this letter. We, thus, encourage you to submit a revised version of the manuscript back to LSA that responds to all of the reviewers' points.

Thank you for this interesting contribution to Life Science Alliance. We are looking forward to receiving your revised manuscript.

Sincerely,

B. MANUSCRIPT ORGANIZATION AND FORMATTING:

Reviewer #1 (Comments to the Authors (Required)):

Cancer treatments such as radiotherapy or chemotherapy have been proposed to promote immunogenic cells death resulting in the release of death-associated molecular patterns acting as adjuvants, promoting DC maturation leading ultimately to T cell priming. The current manuscript by Blair et al. addresses the consequences of radiotherapy on the behaviour (mainly migration properties) and maturation of Dendritic Cells (DCs). Using immunogenic and poorly immunogenic tumor models the authors argue that radiotherapy may result in contrasting effects on the properties of migratory DC (mainly homing to the LN).

The article is well written but some concerns with the experimental and or analytical approaches exist in the current manuscript and should be addressed.

Major points:

1- In its current form, the manuscript suggests that Kaede Red+ migratory cDCs found in the tumor draining LN all originate from the tumor. In order to substantiate the origin of the photoconverted migratory cDCs, the authors should provide additional evidence measuring the extent of photoconversion at the tumor site and nearby tissue (unexposed to UV light). These data should strengthen the authors conclusions about the origin of Kaede Red+ cDC1 and cDC2 in the draining LNs. This is an important point as unconverted (Kaede Red-) mDC1s or mDC2s are considered by the author to not originate from the Tumor.

2- The authors suggest that radiotherapy increased the proportion of "mature" migratory cDCs. An alternative explanation could be that radiotherapy promote "survival" of mature cells. This possibility remains to be explored given the observation that mDC1 maturation appears to be independent of radiotherapy (at least for day 1 and 2 Fig 1D).

Again here it will be of interest to show the maturation status of the non-converted mDCs as it would strengthen the current statement made by the authors (lane 274).

MFI should be provided

3-In general the maturation state of mDCs is limited by a limited a markers. Recent evidences point to the complex nature of mDCs (Maier et al Nature 2020, Cheng et al Cell 2021), and therefore a better characterization of the mig DCs in the Kaede mice is warranted.

3- The authors claim that radiotherapy impairs migratory properties of DCs challenged with poorly radioimmunogenic tumor. This certainly appears to be the case with the Panc02-SIY model (Figure 3), but this data is not corroborated when mice are inoculated with the poorly radioimmunogenic Moc2 tumor cell (Supp Fig 2)?

4-scRNAseq analysis does not appear to separate DC1 from DC2 and the recently described mregDC proposed by the group of Merad (Maier et al Nature 2020), Cheng et al Cell 2021 also highlighted the existence of a similar DC population in several human tumors. These populations should be reflected in the UMAP plot as they harbor a very distinctive transcriptome profile. This is important as all the subsequent analysis performed by the authors are done on the whole cDC population? This interferes with the capacity of the authors to extract valuable information from their analysis.

5- Comparison of the transcriptome of tumor infiltrating DCs and pathway analysis should be also performed in absence of radiation. MC38 being highly immunogenic, therefore one can expect that the transcriptome of cDCs isolated from MC38 bearing mice will differ from the transcriptome of cDCs isolated from P2SIY bearing mice (DC maturation signature may be already featured by cDCs isolated from MC38 tumors, independently of RT treatment?)

additional points

- Definition of cDC2 is blurry. The current gating strategy does not exclude Langerhans Cells?

- From a reader perspective, it will be of interest to identify the nature of the immune cells composing each cluster (Fig 4) as it will strengthen the clustering approach.

Reviewer #2 (Comments to the Authors (Required)):

The paper by Blair et al investigates the requirement of DCs to elicit an antitumor immune response following radiotherapy. This concept is understudied in the field and warrants further investigation. The authors use an innovative means to track intratumoral DCs using photoconvertible KAEDE mice to which they have crossed onto other genetic mouse strains that modulate DCs migration. The manuscript confirms the importance of DCs in initiating the RT-induced antitumor immune response. Furthermore, the authors demonstrate that RT increases the number of DCs that migrate from the tumor to the TdLN (conversion from green to red) in radioimmunogenic tumors. Also, the DCs express a more mature phenotype. These observations are reversed in poorly radioimmunogenic tumor models, which can be improved using poly IC (justified by mechanistic scRNA-seq data suggesting involvement of type I IFNs resulting from damage). Overall, the paper is very well written and the conclusions from the manuscript will advance the field forward. The methods used are innovative, however although the authors tout the use of photoconvertible mice as a means to overcome obstacles, this model is not without its own limitations. The manuscript could be improved if the authors discussed how these limitations could influence their interpretations in some of the experiments. This could be done in the text or possibly with inclusion of additional controls.

- The use of the photoconvertible KAEDE mice are key to this manuscript and provide valuable data. However, many key methods are missing along with a thorough discussion to the limitations this model has. These are summarized below:
 1. How soon after photoconversion are the tumors irradiated (minutes, hours?).
 2. The authors expose the tumor area for 5 minutes. Have controls been performed to show just how much the 405 laser will penetrate into the tumor at this duration of time? How much of the tumor is actually photoconverted? Please discuss or show data?
 3. Is the area shaved before photoconversion? Albino BL/6 are often used in this setting-please comment on the potential of the laser being absorbed by the pigmented skin of BL/6 mice.
 4. The authors expose the tumor area (tumors injected SQ) to the 405 laser, but this needs to penetrate the skin before it reaches the tumor. Do the authors know whether the dermal DCs are migrating to the lymph node (especially after RT)? Has a control been performed where the area of skin is photoconverted and red DCs examined for in the draining LN in a naïve non-tumor bearing mouse? This may give the reader a sense as to the basal migration of DCs to the LN in a non-tumor setting (there likely will be some due to the high 12Gy dose). Please comment.
 5. In figure 3, the authors conclude that RT impairs DC migration to the TdLN since they find less red converted cells on day 3. Have the authors performed a time course (D1, D2, D3) in this model similar to figure 1 (MC38 model). If not, perhaps the migration was missed. Please comment. Also, converted cells will not retain RFP indefinitely. There will be an eventual loss of red. Did the authors do controls showing how long the RFP is retained after conversion? Could the DCs in the Pan02 model be metabolically different and thus have an altered RFP half-life? Please comment.
- In figure 5, the authors treat the poorly radioimmunogenic cell line with RT+ poly IC and witness a modulation of the number of migrated DCs. However, the authors fail to show data whether this has an improved antitumor effect? There is one small comment in the last paragraph in the discussion, however it would help if this was expanded a bit (even if there was no antitumor effect-stating a potential reason for a lack of response would be helpful (and show transparency) to the reader).

Reviewer #3 (Comments to the Authors (Required)):

This manuscript by Blair et al seeks to characterize the effects of radiation on tumor-associated dendritic cell (DC) maturation and trafficking to the tumor draining lymph nodes (TdLN). Specifically, they compare in detail the highly radioimmunogenic transplant tumor model MC38 with the poorly radioimmunogenic model panc02-SIY. They conduct experiments in the photoreactive Kaede mouse model to differentiate cells that were in the tumor from those that were in non-tumor tissue also draining to the TdLN. Using these models, the authors successfully demonstrate significant increases in mature tumor DCs in the TdLN following radiation therapy in the MC38 model, but not in the panc02-SIY model. Furthermore, they conduct RNA-sequencing experiments with pathway analyses to predict candidate effector molecules that contribute to this differential in DC maturation between the models. Taken together, while the findings of this manuscript are limited to only a few transplanted tumor models, it still provides important insights into how radiotherapy can directly alter DC trafficking and maturation, specifically through CCR7.

Major concerns:

The Kaede mouse system is used extensively in this paper, but important controls for the system in the setting of the employed tumor models are not shown. Please include a model validation figure in the supplemental section demonstrating the proportion of tumor cells that are photoactivated from each of the various tumor models. It is unclear how deep the photoconversion penetrates into a mass and it could vary in the different models.

On line 414, the authors describe using the Upstream Regulator analysis for immune based pathways in IPA and they depict the results in Figure 4G. It would strengthen the manuscript to include an additional validation experiment of these results in the two tumor models with and without radiation, particularly for the predicted cytokine pathways. Perhaps an analysis of the total CD45+ RNA-seq transcripts for cytokine production would demonstrate disparate expression levels for the candidate cytokines.

Alternatively, tumor lysate ELISAs to quantify the cytokine milieu would be another approach to strengthen these results.

On line 358, the authors include the MOC1 and MOC2 tumor models in an attempt to broaden their findings. The authors mention that the radioimmunogenic MOC1 model demonstrates similar DC changes following radiation as the MC38 model. However, they fail to state that the poorly radioimmunogenic MOC2 model also demonstrates similar DC changes following radiation as the MC38 model (as seen in Figure S2). This should be clarified and stated clearly in the results. Additionally, given the low magnitudes of change, an expansion cohort of animals could enhance the resolution of these results. If further study continues to show that the poorly radioimmunogenic MOC2 model has DC changes, then the authors need to transparently state that other mechanisms of resistance are the cause of the poor immunogenicity in the MOC2 model.

The authors demonstrated the effects of exogenous poly I:C on the radiation response of panc02-SIY tumors. Adding this exogenous poly I:C experiment in the MC38 model is important so that the readers have a comparison group to better understand the magnitude of effect by the poly I:C in a poorly vs highly radioimmunogenic tumor model.

As recently published by Wisdom et al in Nature Communications in 2020, there are major differences between immune infiltrates (both myeloid and lymphoid compartments) of autochthonous and transplanted tumors. This should be considered in the discussion when describing potential limitations of this study as the observed radiation-associated tumoral DC maturation and trafficking may be different in autochthonous tumors.

Minor comments:

Figures 1 and 3 can be challenging to compare given the differences in the plot styles. While this is understandable due to the additional timepoints in the MC38 model, it would simplify the comparison for readers if the full time course in Figure 1 was moved to the supplemental figures and only the day 3 timepoint was used in Figure 1.

In Figure 4, there are 2 "F" headings instead of the last one being "G".

In the response below the reviewer points are in **bold**, our response in plain text, and changes to the manuscript are underscored. The references in this response letter are consistent, but the numbering will be different in the final manuscript where there are additional references and the order is different.

Reviewer #1 (Comments to the Authors (Required)):

Cancer treatments such as radiotherapy or chemotherapy have been proposed to promote immunogenic cells death resulting in the release of death-associated molecular patterns acting as adjuvants, promoting DC maturation leading ultimately to T cell priming. The current manuscript by Blair et al. addresses the consequences of radiotherapy on the behaviour (mainly migration properties) and maturation of Dendritic Cells (DCs). Using immunogenic and poorly immunogenic tumor models the authors argue that radiotherapy may result in contrasting effects on the properties of migratory DC (mainly homing to the LN).

The article is well written but some concerns with the experimental and or analytical approaches exist in the current manuscript and should be addressed.

Major points:

1- In its current form, the manuscript suggests that Kaede Red+ migratory cDCs found in the tumor draining LN all originate from the tumor. In order to substantiate the origin of the photoconverted migratory cDCs, the authors should provide additional evidence measuring the extent of photoconversion at the tumor site and nearby tissue

(unexposed to UV light). These data should strengthen the authors conclusions about the origin of Kaede Red+ cDC1 and cDC2 in the draining LNs.

This is an important point as unconverted (Kaede Red-) mDC1s or m"DC2s" are considered by the author to not originate from the Tumor.

The reviewers are correct, and this is an important point to the manuscript. We focus on the converted as these are the only cells that we can confirm came from the UV-exposed field. Not all of the cells in the UV-exposed tumor are converted, due to depth of penetrance effects, and due to random inefficiencies. However, the majority of the cells in the TDLN are not converted, and as the reviewer mentions, we do not know the origin of these cells. To address the consequence of unexposed skin, we have performed additional analyses and added a supplementary figure. We show that lymph nodes draining skin that has not been exposed to UV do not have converted dendritic cells (**Supplemental Figure 2**). These data confirm that conversion is dependent on exposure to UV, as with our tumor analyses. In addition, we have performed photoconversion on tumor-free skin, and characterized the photoconverted cells in the draining lymph node. As with the tumor draining lymph node, we only see significant proportions of converted cells in the migratory populations (migratory CD103+ and migratory CD11b+) (**Supplemental Figure 2**). These data confirm that skin DC can be photoconverted and drain to the lymph node, as has been shown by others. This means that as we have stated, we cannot be certain that the converted DC come from the tumor and not from the overlying skin, only that they come from the UV treatment field.

To address the extent of photoconversion in the tumor, we have performed additional analyses and added a supplementary figure that demonstrates that up to 80% of myeloid cells in the tumor are photoconverted immediately following UV exposure (**Supplemental Figure 3**). This is important since in this analysis the tumor is excised from the overlying skin before preparing the single cell suspension. These data suggest a very high rate of photoconversion in the tumor despite the limitations of UV penetration. We then follow the rate of conversion over time in the tumor, and we see that more stable populations such as TAM retain a significant converted population 3 days following conversion (**Supplemental Figure 3**). Neutrophils, which are very short lived, show a steady loss of converted cells in the tumor. Interestingly, monocytes, which are likely to differentiate into different cell types have an extremely short period of conversion and are likely replaced rapidly by newly recruited monocytes that are not converted. The DC populations show a decline in converted cells over time, more similar to neutrophils than macrophages or monocytes. In each cell type we cannot detect a consistent effect of radiation therapy on the proportion converted, suggesting that radiation is not impacting turnover of the Kaede protein.

To describe this, we have added the following to the manuscript:

"Strikingly, we noted that within the TdLN, the majority of converted Kaede-Red⁺ DCs were migratory CD103⁺ cDC1 and CD11b⁺ cDC2 subsets, validating that these were indeed migratory populations (**Figure 1B**). Further confirming the specificity of this model, we did not find any converted DCs in the contralateral inguinal LN (**Figure 1B**). To compare the effect of photoconversion of skin in the absence of a tumor, we analyzed the lymph node draining unconverted skin versus photoconverted skin. One day following conversion, we find that as with the tumor draining lymph node, we only see

significant proportions of converted cells in the migratory CD103⁺ cDC1 and CD11b⁺ cDC2 subsets (Figure S2). To address the extent of photoconversion in the tumor, we analyzed the photoconversion of myeloid populations in the tumor immediately following photoconversion and over time, in the presence and absence of radiation therapy. We demonstrate that up to 80% of myeloid cells in the tumor are photoconverted immediately following UV exposure (Figure S3). When we follow photoconversion over time, we find that more stable populations such as F4/80⁺ Ly6C⁻ MHCII⁺ TAM retain a significant converted population 3 days following conversion (Figure S3). Neutrophils, which are very short lived, show a steady loss of converted cells in the tumor. Interestingly, monocytes, which are likely to differentiate into different cell types have an extremely short period of conversion (Figure S3) and are likely replaced rapidly by newly recruited monocytes that are not converted. The DC populations show a decline in converted cells over time, more similar to neutrophils than macrophages or monocytes. In each cell type we cannot detect a consistent effect of radiation therapy on the proportion of converted cells in the tumor (Figure S3), suggesting that radiation is not directly impacting turnover of Kaede protein in the tumor myeloid populations. Taken together these data suggest that the Kaede mice are a useful tool to study DC populations in the TdLN that have directly migrated from the conversion field following treatment with radiation therapy."

2- The authors suggest that radiotherapy increased the proportion of "mature" migratory cDCs. An alternative explanation could be that radiotherapy promote "survival" of mature cells. This possibility remains to be explored given the observation that mDC1 maturation appears to be independent of radiotherapy (at least for day 1 and 2 Fig 1D).

Again here it will be of interest to show the maturation status of the non-converted mDCs as it would strengthen the current statement made by the authors (lane 274). MFI should be provided.

This is a good point, and it is possible that immature dendritic cells and mature dendritic cells may be differentially impacted by radiation toxicities. However, we don't see this reflected in the absolute number of converted dendritic cells arriving in the lymph node in the MC38 model. As discussed above, when we look at the residual converted cells in the tumor over time, we do not see evidence of differential survival of myeloid populations following radiation therapy (Supplemental Figure 3). These cells are not significantly changed in number following radiation, but their maturation markers increase.

It is a very good idea of the reviewer' to examine the unconverted population. These may include unconverted DC from the tumor as well as DC from non-tumor regions in the lymphatic basin, therefore they represent a good internal control. To address this, we made use of the new analysis of DC maturation in MC38 tumors treated with radiation therapy and pIC that has also been requested by review. When we gate on the converted and the unconverted separately, then analyze the maturation marker CD80 in the DC populations, we can see that in untreated tumors CD80 is not significantly different between converted and unconverted cells (Supplemental Figure 11). However, following radiation and pIC treatment, there is a dramatic increase in CD80 that is almost entirely confined to the converted population. These data suggest that the effect of tumor treatment dominantly occurs in the converted population.

To address this, we have added the following to the manuscript:

“These data support the hypothesis that the innate adjuvant balance in tumors is critical to support adaptive immune responses following radiation therapy¹, but also demonstrate that cancer cells can establish distinct environments in genetically identical mice that can suppress innate and therefore adaptive immunity to tumors. To understand the effects of the tumor environment on DC that migrated from the tumor versus the effects that both converted and unconverted DC may experience in the TDLN, we analyzed the maturation status of converted versus unconverted migratory CD103⁺ DC and CD11b⁺ DCs in the TdLN of MC38 tumors treated with radiation therapy and pIC. In untreated tumors CD80 is not significantly different between converted and unconverted DCs (Figure S11). However, following radiation and pIC treatment, there is a dramatic increase in CD80 that is almost entirely confined to the converted DC populations. These data suggest that the effect of tumor treatment dominantly occurs in the converted population, and is unlikely to be occurring in the TdLN where unconverted DC would receive an equal maturation signal.”

3-In general the maturation state of mDCs is limited by a limited a markers. Recent evidences point to the complex nature of mDCs (Maier et al Nature 2020, Cheng et al Cell 2021), and therefore a better characterization of the mig DCs in the Kaede mice is warranted.

Unfortunately, using the Kaede mouse has significantly limited our ability to profile DC. While our flow cytometer has 15 available colors, the very bright 'green' and 'red' signals from the mice leaches into additional channels regardless of our attempts to compensate, forcing us to avoid overlapping fluorescence channels. In addition, given the need for many markers to identify the DC subtypes (CD45, live/dead, CD90, CD19, CD11c, Ly6C, MHCII, CD8a, and CD103), plus Kaede Red and Kaede green, we only have space for CD40 and CD80 as maturation markers.

However, our profiling antibodies broadly overlap with those described to identify DC in the Maier et al Nature 2020 paper, and in this paper CD40 and CD80 were similarly markers of activation associated with antigen uptake in the tumor, and antigen transport to the TDLN. The Cheng et al 2021 Cell paper that uses scRNASeq to profile DC, Clec9a is a major marker of the DC1 population that distinguishes these cells from the Lamp3+ DC population. We similarly show using scRNASeq that the DC1 population we focus on express Clec9a (Figure 4). The Cheng et al paper links the Lamp3+ population (which can come from DC1 or DC2 origins) to the MregDC population in the Maier et al paper, suggesting that based on their transcriptional signal, these may be the same cells. However, Cheng et al focus on human DC, and when we profiled the murine DC we were unable to detect Lamp3 expression in any population (these data are discussed more in a later review question). The combination of our phenotyping data and transcriptional data suggest that in reference to these papers, our Kaede+ cDC1 cells are more likely mature cDC1 that overlap more with MregDC than Lamp3+ DC.

To address this in the manuscript, we have added the following:

“Starting 2 days following treatment there was a significant increase in the proportion of converted tumor migratory CD103⁺ cDC1s co-expressing CD40 and CD80 in the radiation treated group compared to untreated controls (**Figure 1D ii**). The strong fluorescence of the Kaede transgene limited the number of maturation markers we could study on the DC in the TdLN; however, CD80 and CD40 are relevant and robust measures of dendritic cell maturation in lymph nodes². Thus, broadly the DCs from radiation treated animals were more mature than the tumor migratory DC populations in the TdLN from untreated controls.”

3- The authors claim that radiotherapy impairs migratory properties of DCs challenged with poorly radioimmunogenic tumor. This certainly appears to be the case with the Panc02-SIY model (Figure 3), but this data is not corroborated when mice are inoculated with the poorly radioimmunogenic Moc2 tumor cell (Supp Fig 2)?

This is a good point. There are likely multiple mechanisms at work here. Moc2 tumors have an almost total absence of CD8 T cells infiltrating the tumor, and this appears to be due to extremely poor inflammation-based recruitment of T cells to the tumor^{3,5}. For this reason, even if Moc2 were able to functionally cross-present antigen in the TdLN, the tumor may remain ignorant. This is consistent with our prior experience following radiation therapy of Moc2 – these tumors are unresponsive to radiation and T cells play no role in tumor control following radiation in this model⁶. To address this directly, we have added the following to the manuscript:

“To ensure that these were not model-specific events, we assessed DC migration following radiation therapy in two additional head and neck tumor models, radioimmunogenic Moc1 and poorly radioimmunogenic Moc2 tumors (**Figure S5A**). While the number of converted tumor migratory CD103⁺ DCs and CD11b⁺ did not change in either group, there was an increased proportion of both DC subsets co-expressing CD40 and CD80 in the radiation treated Moc1 group (**Figure S5B i-ii, C i-ii**). While Moc2 is a poorly radioimmunogenic tumor^{6,7}, Moc2 tumors exhibited an increase in CD103⁺ DC but not CD11b⁺ DC co-expressing CD40 and CD80 in the TdLN following radiation therapy (Figure S5B i-ii, C i-ii). Given that this tumor has an extremely poor infiltrate of T cells in the tumor⁸, it is possible that the mechanisms driving poor radioimmunogenicity may not be related to DC maturation in this tumor model⁷. Thus, in the well-infiltrated Panc02-SIY model, treatment with radiation reduces the number of tumor migratory cDCs that are found in the TdLN and this may explain why the Panc02-SIY tumor model is poorly responsive to radiation therapy.”

4-scRNAseq analysis does not appear to separate DC1 from DC2 and the recently described mregDC proposed by the group of Merad (Maier et al Nature 2020), Cheng et al Cell 2021 also highlighted the existence of a similar DC population in several human tumors. These populations should be reflected in the UMAP plot as they harbor a very distinctive transcriptome profile. This is important as all the subsequent analysis performed by the authors are done on the whole cDC population? This interferes with the capacity of the authors to extract valuable information from their analysis.

As discussed above, the transcriptional profile of the DC we analyze by scRNASeq overlaps best with cDC1. All of the key markers proposed by Maier et al and Cheng et al are present (Clec9a, Xcr1, Itgae (CD103)), but these also have some maturation features (Ccr7, Il12b), which places the cells closer to the MregDC proposed by Dr. Merad's group ², and the Clec9a+ cDC1 population proposed by Dr. Cheng et al. ⁹ We do not detect Lamp3 expression in these cells, which excludes these being the Lamp3 population that can arise from DC1 or DC2 origin, though this may relate to the fact that we are analyzing cells from murine tumors. However, our focus on these cells was centered on the population identified as DC by graph-based clustering and found to enrich for these key genes, but also cells that were expressing the Zbtb46 gene. This is critical since these are the cells that we deplete using the Zbtb46-DTR mice, that we show are critical for responsiveness to radiation therapy, and must express CCR7 to mediate that effect.

As the reviewers suggest, it is possible that we have a mixed population in our DC cluster. To address this question, we have attempted to separate the original cluster 16 population that express DC markers (**Figure 4**) into additional clusters. Initial analysis identified 3 clusters of cells within this population (**Supplementary Figure 8**). We then analyzed the expression of the distinctive genes that separate DC subpopulations in the Maier et al paper ². Using this approach we can identify that cluster 3 has more features of mregDC, with expression of the activation markers Ccr7, Fcsc1, and Relb, while cluster 2 has more features of DC1, with expression of Naaa, Cadm1, and Xcr1 (**Figure S8**) (2). By contrast cluster 1 has features of DC2, with higher expression of Sirpa, H2-DMb2, and Itgam. These data indicate that our cDC population is consistent with other defined DC populations by single cell RNASeq.

To address this, we have added the following to the manuscript:

“To define cDCs for our analysis we used cells that were in cluster 16 and expressed one or more Zbtb46 transcripts (**Figure 4D, S7A**). Recent studies have identified multiple DC populations in human and murine tumors with distinct transcriptional profiles ^{2,9}. To identify these profiles in our tumors, we again used the clustering algorithm from the Loupe Cell Browser to separate the DC into distinct populations (**Figure S8**). We then analyzed the expression of the distinctive genes that separate DC subpopulations in these studies. Using this approach we can identify that cluster 3 has more features of mregDC, with expression of the activation markers Ccr7, Fcsc1, and Relb, while cluster 2 has more features of DC1, with expression of Naaa, Cadm1, and Xcr1 (**Figure S8**) ². By contrast cluster 1 has features of DC2, with higher expression of Sirpa, H2-DMb2, and Itgam. These data indicate that our cDC population is consistent with other defined DC populations by single cell RNASeq. As an initial analysis of the DC in our tumors, we compared differences in gene expression that occur following radiation in cDCs for each tumor type using differential gene expression analysis (**Figure S7B**).”

5- Comparison of the transcriptome of tumor infiltrating DCs and pathway analysis should be also performed in absence of radiation. MC38 being highly immunogenic, therefore one can expect that the transcriptome of cDCs isolated from MC38 bearing mice will differ from the transcriptome of cDCs isolated from P2SIY bearing mice (DC maturation signature may be already featured by cDCs isolated from MC38 tumors, independently of RT

treatment?)

This is a good point, and there are differences in the DC between these tumors at baseline. This is likely important since we would expect these differences to be in place prior to radiation so that the differences in maturation are revealed on radiation treatment. To address this, we have added a supplemental figure that profiles the differences in gene expression between these cells at baseline and on maturation.

We have added the following to the manuscript“

“As an initial analysis of the DC in our tumors, we compared differences in gene expression that occur between the DC infiltrating different tumors at baseline, and following radiation therapy (Figure S7A-B). Of the 15 genes that were upregulated following radiation therapy in both tumors, these mapped to the cellular response to radiation (Cdkn1a), interferon gamma (Cd274), and components of the S100 family (Figure S7C). To focus on the differences in gene expression that occur following radiation in cDCs for each tumor type (Figure S7B) using differential gene expression analysis, we identified that in MC38 cDCs there were 93 genes that were upregulated with radiation and 116 genes in Panc02-SIY cDCs (Figure S7B).”

- Definition of cDC2 is blurry. The current gating strategy does not exclude Langerhans Cells?

At present we are unable to exclude Langerhans cells from the various populations. However, the consensus definition described by Merad et al 2013 Annu Rev Immunol (<https://www.ncbi.nlm.nih.gov/pmc/articles/PMC3853342/>) and Kaplan 2017 Nature Immunology (<https://www.ncbi.nlm.nih.gov/pmc/articles/PMC6422157/>) suggests that Langerhans cells can express F4/80, CD11b, and do not express CD103, so would not be included in the DC populations we see in the tumor, nor in the CD103+ cDC populations that we see trafficking from the tumors.

To address this in the manuscript, we have added the following:

“Given that the overlaying skin is included in the radiation field and the field of UV conversion, it is possible that alternative dendritic cell populations such as Langerhans cells are among the converted cells in the TDLN. However, Langerhans cells in the lymph node are defined as F4/80⁺CD11b⁺CD103⁻^{10,11}, therefore these are not among the CD103⁺ DC population that differentially mature following radiation therapy.”

- From a reader perspective, it will be of interest to identify the nature of the immune cells composing each cluster (Fig 4) as it will strengthen the clustering approach.

This is a good point, since while we have focused on the DC, there are many other populations. We have added a supplemental figure that identifies key immune subtypes. These now shows the dominant markers in the infiltrating cells, specifically the T cell, myeloid, and NK cell clusters according to expression of key markers such as Cd3e, Csf1r, etc.

We have added the following to the manuscript:

“Using the graph-based cluster algorithm, 18 clusters were identified, with cluster 16 expressing many cDC related genes such as Flt3, Itgae, Tlr3 and Zbtb46. **(Figure 4B-C)**. These cDC populations were distinct from markers denoting other key immune populations across the tumor infiltrating cells (Figure S6). To define cDCs for our analysis we used cells that were in cluster 16 and expressed one or more Zbtb46 transcripts **(Figure 4D, S7A)**.”

Reviewer #2 (Comments to the Authors (Required)):

The paper by Blair et al investigates the requirement of DCs to elicit an antitumor immune response following radiotherapy. This concept is understudied in the field and warrants further investigation. The authors use an innovative means to track intratumoral DCs using photoconvertible KAEDE mice to which they have crossed onto other genetic mouse strains that modulate DCs migration. The manuscript confirms the importance of DCs in initiating the RT-induced antitumor immune response. Furthermore, the authors demonstrate that RT increases the number of DCs that migrate from the tumor to the TdLN (conversion from green to red) in radioimmunogenic tumors. Also, the DCs express a more mature phenotype. These observations are reversed in poorly radioimmunogenic tumor models, which can be improved using poly IC (justified by mechanistic scRNA-seq data suggesting involvement of type I IFNs resulting from damage). Overall, the paper is very well written and the conclusions from the manuscript will advance the field forward. The methods used are innovative, however although the authors tout the use of photoconvertible mice as a means to overcome obstacles, this model is not without its own limitations. The manuscript could be improved if the authors discussed how these limitations could influence their interpretations in some of the experiments. This could be done in the text or possibly with inclusion of additional controls.

• The use of the photoconvertible KAEDE mice are key to this manuscript and provide valuable data. However, many key methods are missing along with a thorough discussion to the limitations this model has. These are summarized below:

1. How soon after photoconversion are the tumors irradiated (minutes, hours?).

The reviewer is correct, this is important information. At most there is an hour lapse between photoconversion and radiation therapy. To address this, we have added a line to the materials and methods as follows:

“Briefly, animals were completely covered in aluminum foil except for tumors which were exposed to 405nm LED light source using a collimator for 5 minutes (Prizmatix). Where radiation therapy followed Kaede conversion, the UV conversion in Kaede mice was performed immediately prior to radiation therapy, with at most a 1 hour lapse between conversion and radiation treatment.”

2. The authors expose the tumor area for 5 minutes. Have controls been performed to show just how much the 405 laser will penetrate into the tumor at this duration of time? How much of the tumor is actually photoconverted? Please discuss or show data?

Penetration is imperfect with the light source. As discussed with reviewer 1, we typically see photoconversion of approximately 80% of cells in the tumor. This likely results from limited depth of penetration, combined with random

factors for individual cells. This was nicely demonstrated in a paper by Torcellan *et al*¹², which showed decreasing conversion with increasing distance from the skin.

To discuss this, we have added the following to the manuscript:

“To address the extent of photoconversion in the tumor, we analyzed the photoconversion of myeloid populations in the tumor immediately following photoconversion and over time, in the presence and absence of radiation therapy. We demonstrate that up to 80% of myeloid cells in the tumor are photoconverted immediately following UV exposure (Figure S3).”

3. Is the area shaved before photoconversion? Albino BL/6 are often used in this setting-please comment on the potential of the laser being absorbed by the pigmented skin of BL/6 mice.

These mice are shaved, but are not albino. Given the relatively high proportion of cells being converted, we opted not to backcross the Kaede transgene onto Albino mice. This is the same system used by our co-author and collaborator Dr. Lund in her recent work (<https://pubmed.ncbi.nlm.nih.gov/30663703/>).

To clarify this, we have added the following to the methods:

“For photoconversion experiments using the Kaede mice, tumors were converted as described by Steele *et al*¹³. Briefly, tumors were implanted in animals with shaved skin, and for photoconversion animals were completely covered in aluminum foil except for the tumors which were exposed to 405nm LED light source using a collimator for 5 minutes (Prizmatix).”

4. The authors expose the tumor area (tumors injected SQ) to the 405 laser, but this needs to penetrate the skin before it reaches the tumor. Do the authors know whether the dermal DCs are migrating to the lymph node (especially after RT)? Has a control been performed where the area of skin is photoconverted and red DCs examined for in the draining LN in a naïve non-tumor bearing mouse? This may give the reader a sense as to the basal migration of DCs to the LN in a non-tumor setting (there likely will be some due to the high 12Gy dose). Please comment.

This is an important point. The dermal DC will be photoconverted along with those in the tumor beneath. As discussed, the tumor dominates due to the absolute number of cells in the tumor being much higher than in the overlaying skin. However, Tomura *et al* (<https://www.pnas.org/content/105/31/10871>) and our co-author Dr. Lund (<https://pubmed.ncbi.nlm.nih.gov/30663703/>) have shown that DC from photoconverted skin do traffic to the draining lymph node, and so would be among our analyzed cells. To address the number of such cells we performed an additional experiment as discussed with regards to reviewer 1’s comments above. We took matched mice and converted tumor-free skin one group and left the other unconverted. 1 day later we harvested the lymph node draining

the conversion site, and profiled the DC in the LN. As with the TDLN, in the skin DLN converted cells were only found in the migratory CD103⁺ DC and the migratory CD11b⁺ DC (**Supplementary Figure 2**). No converted cells were found in the lymph node that had not been exposed to UV. As discussed with Reviewer 1 above, we cannot distinguish whether the DC in the TDLN originated in the tumor or in the overlaying skin. However, it is of interest that when we directly inject pIC into the tumor then profile the maturation status of the converted versus unconverted DC in the TDLN, we see that only the converted cells are matured (**Supplementary Figure 11**). While the tumor is growing in the subcutaneous space, there is likely no interchange of pIC from the tumor, through the capsule, to the dermis. We see very small proportions of converted DC that have not matured following treatment, and very little evidence of unconverted DC maturation following treatment (**Supplementary Figure 11**). These data suggest that dermal DC represent a very small proportion of the converted DC in the TDLN. Despite this, they cannot be excluded from the analysis.

To address this in the manuscript, we have added the following:

“To compare the effect of photoconversion of skin in the absence of a tumor, we analyzed the lymph node draining unconverted skin versus photoconverted skin. One day following conversion, we find that as with the tumor draining lymph node, we only see significant proportions of converted cells in the migratory CD103⁺ cDC1 and CD11b⁺ cDC2 subsets (**Figure S2**).”

5. In figure 3, the authors conclude that RT impairs DC migration to the TdLN since they find less red converted cells on day 3. Have the authors performed a time course (D1, D2, D3) in this model similar to figure 1 (MC38 model). If not, perhaps the migration was missed. Please comment. Also, converted cells will not retain RFP indefinitely. There will be an eventual loss of red. Did the authors do controls showing how long the RFP is retained after conversion? Could the DCs in the Pano2 model be metabolically different and thus have an altered RFP half-life? Please comment.

We have performed a time course in the Pano2 model, and found that cell emigration from the tumor is quite stable over time. There does not seem to be an earlier emigration in the Pano2 model; however, there could yet be a later emigration that we have not identified as we have not profiled beyond day 3 in these models. The red fluorescence is very stable in converted cells. Tomura et al demonstrated no loss of fluorescence at 7 days following conversion in one of the original papers describing the system (<https://www.pnas.org/content/105/31/10871>). We have similarly seen no loss of fluorescence by 3 days following conversion, though highly proliferative cells can proportionately lose red fluorescence over time as they divide. However, we and others have seen no evidence of proliferation in fully matured dendritic cells. As discussed with Reviewer 1, we have profiled the timeline of myeloid cell conversion in the two tumor types over time, and we see a very similar pattern of conversion and loss of converted cells over time between the tumors (**Supplementary Figure 3**).

To address this in the manuscript, we have added the following:

“To address the extent of photoconversion in the tumor, we analyzed the photoconversion of myeloid populations in the tumor immediately following photoconversion and over time, in the presence and absence of radiation therapy. We demonstrate that up to 80% of myeloid cells in the tumor are photoconverted immediately following UV exposure (Figure S3). When we follow photoconversion over time, we find that more stable populations such as F4/80⁺ Ly6C⁻ MHCII⁺ TAM retain a significant converted population 3 days following conversion (Figure S3). Neutrophils, which are very short lived, show a steady loss of converted cells in the tumor. Interestingly, monocytes, which are likely to differentiate into different cell types have an extremely short period of conversion (Figure S3) and are likely replaced rapidly by newly recruited monocytes that are not converted. The DC populations show a decline in converted cells over time, more similar to neutrophils than macrophages or monocytes. In each cell type we cannot detect a consistent effect of radiation therapy on the proportion of converted cells in the tumor (Figure S3), suggesting that radiation is not directly impacting turnover of Kaede protein in the tumor myeloid populations.”

• In figure 5, the authors treat the poorly radioimmunogenic cell line with RT+ poly IC and witness a modulation of the number of migrated DCs. However, the authors fail to show data whether this has an improved antitumor effect? There is one small comment in the last paragraph in the discussion, however it would help if this was expanded a bit (even if there was no antitumor effect-stating a potential reason for a lack of response would be helpful (and show transparency) to the reader).

This is a good point. We have performed these experiments as part of our prior work where we identified that the combination of RT and polyIC was necessary to generate tumor cures in the Panco2 model⁶. We did not want to re-show this data in the current manuscript, but we should have described it better.

To address this in the manuscript, we have added the following:

“In addition, it is unclear whether there is active suppression of DC maturation in poorly radioimmunogenic tumors, an absence of positive maturation signals, or some combination of both. There are many potential mechanisms that could be responsible for the suppression of DC maturation following treatment, whether coming directly from the tumor or indirectly through differential intratumoral immune populations, for example macrophages¹⁴⁻¹⁷. These regulatory features may include metabolic features of the tumor – for instance, it has been reported that oxidized cholesterol ligands secreted from tumors can suppress the expression of CCR7 in DCs and this impairs the migration from the tumor to TdLN¹⁸. However, the exogenous delivery of maturation stimuli by delivery of adjuvants such as pIC⁶ or STING ligands¹⁹ can improve T cell control of tumors following radiation therapy, suggesting that if active suppression of DC maturation is in place, it can be overcome. Taken together, these data suggest that individual tumor microenvironments determine whether DCs are activated or suppressed following treatment with radiation, and this in turn determines whether DCs migrate to the TdLN to cross-present antigens.”

Reviewer #3 (Comments to the Authors (Required)):

This manuscript by Blair et al seeks to characterize the effects of radiation on tumor-associated dendritic cell (DC) maturation and trafficking to the tumor draining lymph nodes (TdLN). Specifically, they compare in detail the highly radioimmunogenic transplant tumor model MC38 with the poorly radioimmunogenic model panco2-SIY. They conduct experiments in the photoreactive Kaede mouse model to differentiate cells that were in the tumor from those that were in non-tumor tissue also draining to the TdLN. Using these models, the authors successfully demonstrate significant increases in mature tumor DCs in the TdLN following radiation therapy in the MC38 model, but not in the panco2-SIY model. Furthermore, they conduct RNA-sequencing experiments with pathway analyses to predict candidate effector molecules that contribute to this differential in DC maturation between the models. Taken together, while the findings of this manuscript are limited to only a few transplanted tumor models, it still provides important insights into how radiotherapy can directly alter DC trafficking and maturation, specifically through CCR7.

Major concerns:

The Kaede mouse system is used extensively in this paper, but important controls for the system in the setting of the employed tumor models are not shown. Please include a model validation figure in the supplemental section demonstrating the proportion of tumor cells that are photoactivated from each of the various tumor models. It is unclear how deep the photoconversion penetrates into a mass and it could vary in the different models.

This is an important point and was similarly raised by reviewers 1 and 2. We have now added data that demonstrates the proportion of cells that are photoconverted in the models (**Supplemental Figure 3**). These data show that up to 80% of cells in the tumor are photoconverted at baseline, and this data matches well with that shown by other investigators in similar systems. Importantly, since we focus on the converted cells, while not all of the cells may be converted we are confident that the converted cells originated in the treatment field. Any unconverted cells in the tumor environment are not part of the lymph node analysis, so if there are differences between models that are not reflected in the tumor flow cytometry, we are not impacted by the unconverted populations.

To address this in the manuscript, we have added the following:

“Importantly, these changes could not be identified in unconverted DC, and were not detectable when assessing the total DC population of the TdLN (data not shown), emphasizing the importance of the Kaede model in studying these site-specific effects. In addition, while a range of inefficiencies mean that not all of the cells in the tumor are converted, by focusing only on converted cells we ensure that we can be confident that these cells originated in the UV and radiation treatment fields.”

On line 414, the authors describe using the Upstream Regulator analysis for immune based pathways in IPA and they depict the results in Figure 4G. It would strengthen the manuscript to include an additional validation experiment of these results in the two tumor models with and without radiation, particularly for the predicted cytokine pathways. Perhaps an analysis of the total CD45+ RNA-seq transcripts for cytokine production would demonstrate disparate expression levels for the candidate cytokines. Alternatively, tumor lysate ELISAs to quantify the cytokine milieu would be another approach to strengthen these results.

As the reviewer suggested, we have profiled the scRNASeq for candidate mediators identified by IPA analysis at the d1 timepoint. To do this, we analyzed all scRNASeq transcripts for significant changes between irradiated Panc02-SIY versus irradiated MC38 tumors as suggested by the reviewer (Table S1). Within these genes we highlighted those identified as candidate upstream mediators by IPA (Figure S9). Within these genes we see a mixed pattern. Some, such as Spp1, Irf7, and Il1b are upregulated in MC38 tumors as suggested by IPA analysis and may be candidate mediators of altered DC maturation. By contrast others such as Ifng and Il17a are downregulated, suggesting that they are not candidate mediators of altered DC maturation predicted by IPA analysis. However, we must acknowledge that the levels of these transcripts across the total dataset may not be impactful to the DC microenvironment, so we must be cautious in interpreting these data.

Notably, we could not see significant changes in type I IFN genes following radiation therapy. While it remains possible that we do not see these changes since we only profile the CD45+ infiltrating immune cells, MC38 tumors have been shown to upregulate type I IFN 3 days following radiation therapy, and this is dependent on STING expression in host cells²⁰. In this same manuscript the authors demonstrated that macrophages and dendritic cells were the major source of type I IFN transcripts in these cells²⁰. However, the general timeline for STING-related effects are 3-5 days following radiation therapy, and are dependent on the timeline of micronuclei formation in the cancer cells²¹. Given the absence of type I IFN changes in our samples at d1 post RT, and the expected slow timeline of adjuvant activation in the tumor environment, we do not currently have a good explanation for this early maturation effect or RT on DC in MC38 tumors as profiled in the single cell sequencing, though it is more in line with the maturation of converted DC in the tumor draining lymph nodes we see at 3 days following radiation therapy.

To address this in the manuscript, we have added the following:

“These data suggest that innate adjuvant signaling pathways are likely activated in cDCs from MC38 tumors treated with radiation therapy and this leads to successful DC maturation. To determine whether the candidate upstream regulators were altered in the tumor environment, we analyzed all scRNASeq transcripts for significant changes between irradiated Panc02-SIY versus irradiated MC38 tumors (Table S1). Within these genes we highlighted those identified as candidate upstream mediators by IPA (Figure S9). These genes show a mixed pattern. Some candidate genes such as SPP1, IRF7, and IL1B are expressed more in MC38 tumors and may be candidate mediators of altered DC maturation. By contrast others such as IFNG and IL17A are expressed less in MC38 tumors, suggesting that they are not candidate mediators of altered DC maturation that is predicted by IPA analysis. Further studies are necessary to identify the mechanisms resulting in altered DC maturation following radiation therapy.”

And also in the discussion, the following:

“Future studies are needed to perform a comprehensive analysis to identify additional migratory DCs maturation markers so that we can determine whether more subtle difference exist between the migratory DCs. Our scRNASeq analyses show that DC differences in DC maturation can be detected very early following radiation therapy. Notably, we could not see significant changes in type I IFN genes following radiation therapy. While it remains possible that we do not see these changes since we only profile the CD45+ infiltrating immune cells, MC38 tumors have been shown to upregulate type I IFN 3 days following radiation therapy, and this is dependent on STING expression in host cells²⁰, suggesting a host origin for type I IFN transcripts. In this same manuscript the authors demonstrated that macrophages and dendritic cells were the major source of type I IFN transcripts²⁰. STING activation in host cells has been shown to be dependent on the timeline of micronuclei formation in the cancer cells²¹, which can take 3-7 days to generate STING activation in vitro. Given the absence of detectable type I IFN changes in our samples at d1 post RT, and the expected slow timeline of adjuvant activation in the tumor environment, we do not currently have a good explanation for this early maturation effect or RT on DC in MC38 tumors as profiled in the scRNASeq. Importantly, it is unclear whether there is active suppression of DC maturation in poorly radioimmunogenic tumors, an absence of positive maturation signals, or some combination of both.”

On line 358, the authors include the MOC1 and MOC2 tumor models in an attempt to broaden their findings. The authors mention that the radioimmunogenic MOC1 model demonstrates similar DC changes following radiation as the MC38 model. However, they fail to state that the poorly radioimmunogenic MOC2 model also demonstrates similar DC changes following radiation as the MC38 model (as seen in Figure S2). This should be clarified and stated clearly in the results. Additionally, given the low magnitudes of change, an expansion cohort of animals could enhance the resolution of these results. If further study continues to show that the poorly radioimmunogenic MOC2 model has DC changes, then the authors need to transparently state that other mechanisms of resistance are the cause of the poor immunogenicity in the MOC2 model.

This is true, and we do need to address this as requested by reviewer 2. There may be multiple mechanisms at play here. Moc2 is poorly radioimmunogenic as it does not mount a T cell response that contributes to tumor control following radiation therapy⁶, but as the reviewer observes it does seem to permit DC maturation. We believe the failure relates to the almost total absence of T cells in Moc2 tumors³, suggesting that recruitment of any successfully primed T cells does not occur. Moore et al demonstrated that treating Moc2 tumors with inflammatory triggers such as STING ligands are insufficient to control the tumors despite positive effects on the environment⁵, and this may relate to simultaneous negative triggers in the immune environment – for example we needed to block IL-10 for STING ligands to impact Moc2 tumors⁴. To better discuss these data, we have added the following to the manuscript:

“While the number of converted tumor migratory CD103⁺ DCs and CD11b⁺ did not change in either group, there was an increased proportion of both DC subsets co-expressing CD40 and CD80 in the radiation treated Moc1 group (Figure S5B

i-ii, C i-ii). While Moc2 is a poorly radioimmunogenic tumor^{6,7}, Moc2 tumors exhibited an increase in CD103⁺ DC but not CD11b⁺ DC co-expressing CD40 and CD80 in the TdLN following radiation therapy (Figure S5B i-ii, C i-ii). Given that this tumor has an extremely poor infiltrate of T cells in the tumor⁸, it is possible that the mechanisms driving poor radioimmunogenicity may not be related to DC maturation in this tumor model^{4,7}.

The authors demonstrated the effects of exogenous poly I:C on the radiation response of panco2-SIY tumors. Adding this exogenous poly I:C experiment in the MC38 model is important so that the readers have a comparison group to better understand the magnitude of effect by the poly I:C in a poorly vs highly radioimmunogenic tumor model.

This is a good point. If MC38 responds better to polyI:C it may suggest active suppression of DC rather than absence of maturation signals. We have included data demonstrating that polyI:C also matures DC in the TDLN of MC38 tumors (Supplemental Figure 10). We interpret these data to suggest that Panco2 is capable of responding should the appropriate maturation signals be present. To address this, we have added the following to the manuscript:

“When the absolute numbers of tumor migratory DCs were analyzed, we observed significant increases in the total number of Kaede-Red CD103⁺ cDC1 and CD11b⁺ cDC2s (Figure 5C i, D i). Similarly, treatment with poly I:C significantly increased the total number of Kaede-Red CD103⁺ cDC1 and CD11b⁺ cDC2s in the TdLN of MC38 tumors (Figure S10). These data indicate that exogenous administration of poly I:C is capable of driving cDC migration from the tumor to TdLN in both radioimmunogenic and in poorly radioimmunogenic tumors.”

As recently published by Wisdom et al in Nature Communications in 2020, there are major differences between immune infiltrates (both myeloid and lymphoid compartments) of autochthonous and transplanted tumors. This should be considered in the discussion when describing potential limitations of this study as the observed radiation-associated tumoral DC maturation and trafficking may be different in autochthonous tumors.

This is a very good point, in fact, we believe it represents a significant limitation in the use of transplantable tumor models. We have found that implanted tumors result in an immune response to tumor challenge that dramatically impacts subsequent treatments²². Dr. Kirsch's group has very nicely demonstrated that induced tumors develop immune responses very differently. To address this we have added a discussion to the manuscript.

“These data are focused on dendritic cell migration from implanted subcutaneous tumors. However, DC migration from spontaneously emerging tumors may be differently regulated^{23,24}. Moreover, the direct injection of cancer cells into mice can result in a strong immune response that can impact the subsequent immune biology of the tumor^{7,22}, which may be absent in an induced tumor²⁴. For these reasons, it can be critical to explore the dynamics of immunity over time in preclinical models to understand whether they are directly applicable to patient tumors²⁵. In addition, these data suggest that we refocus our attention on the T cells in the TdLN that

recognize the antigens cross-presented by DC, as an alternative to focusing on the exhausted T cell populations that are uniquely found in the tumor environment. "

Minor comments:

Figures 1 and 3 can be challenging to compare given the differences in the plot styles. While this is understandable due to the additional timepoints in the MC38 model, it would simplify the comparison for readers if the full time course in Figure 1 was moved to the supplemental figures and only the day 3 timepoint was used in Figure 1.

This is a difficult one for us. Other reviewers have brought up questions about the time course, so we have added data from different times to the manuscript. However, the direct time-point comparison is important since our manuscript focuses on the differences between MC38 and Panco2. As suggested, we have shown only the d3 data in the manuscript as **Figure 1**, and added the timecourse data to a **Supplemental Figure 4**.

In **Figure 4**, there are 2 "F" headings instead of the last one being "G".

Well spotted! This has been corrected in the current version.

References

1. Baird, J.R., *et al.* Stimulating Innate Immunity to Enhance Radiation Therapy-Induced Tumor Control. *Int J Radiat Oncol Biol Phys* **99**, 362-373 (2017).
2. Maier, B., *et al.* A conserved dendritic-cell regulatory program limits antitumour immunity. *Nature* **580**, 257-262 (2020).
3. Sharon, S., *et al.* Explant Modeling of the Immune Environment of Head and Neck Cancer. *Front Oncol* **11**, 611365 (2021).
4. Baird, J.R., *et al.* Evaluation of Explant Responses to STING Ligands: Personalized Immunosurgical Therapy for Head and Neck Squamous Cell Carcinoma. *Cancer Res* **78**, 6308-6319 (2018).
5. Moore, E., *et al.* Established T Cell-Inflamed Tumors Rejected after Adaptive Resistance Was Reversed by Combination STING Activation and PD-1 Pathway Blockade. *Cancer immunology research* **4**, 1061-1071 (2016).
6. Blair, T.C., *et al.* Dendritic Cell Maturation Defines Immunological Responsiveness of Tumors to Radiation Therapy. *J Immunol* **204**, 3416-3424 (2020).
7. Medler, T.R., Blair, T.C., Crittenden, M.R. & Gough, M.J. Defining Immunogenic and Radioimmunogenic Tumors. *Front Oncol* **11**, 667075 (2021).
8. Sharon, S., *et al.* A platform for locoregional T-cell immunotherapy to control HNSCC recurrence following tumor resection. *Oncotarget* **12**, 1201-1213 (2021).
9. Cheng, S., *et al.* A pan-cancer single-cell transcriptional atlas of tumor infiltrating myeloid cells. *Cell* **184**, 792-809 e723 (2021).
10. Kaplan, D.H. Ontogeny and function of murine epidermal Langerhans cells. *Nat Immunol* **18**, 1068-1075 (2017).

11. Merad, M., Sathe, P., Helft, J., Miller, J. & Mortha, A. The dendritic cell lineage: ontogeny and function of dendritic cells and their subsets in the steady state and the inflamed setting. *Annu Rev Immunol* **31**, 563-604 (2013).
12. Torcellan, T., et al. In vivo photolabeling of tumor-infiltrating cells reveals highly regulated egress of T-cell subsets from tumors. *Proc Natl Acad Sci U S A* **114**, 5677-5682 (2017).
13. Steele, M.M., et al. Quantifying Leukocyte Egress via Lymphatic Vessels from Murine Skin and Tumors. *J Vis Exp* (2019).
14. Ruffell, B., et al. Macrophage IL-10 blocks CD8+ T cell-dependent responses to chemotherapy by suppressing IL-12 expression in intratumoral dendritic cells. *Cancer Cell* **26**, 623-637 (2014).
15. Medler, T., et al. Activating the Nucleic Acid-Sensing Machinery for Anticancer Immunity. *Int Rev Cell Mol Biol* **344**, 173-214 (2019).
16. Tormoen, G.W., Crittenden, M.R. & Gough, M.J. Role of the immunosuppressive microenvironment in immunotherapy. *Adv Radiat Oncol* **3**, 520-526 (2018).
17. Jhunjhunwala, S., Hammer, C. & Delamarre, L. Antigen presentation in cancer: insights into tumour immunogenicity and immune evasion. *Nat Rev Cancer* **21**, 298-312 (2021).
18. Villablanca, E.J., et al. Tumor-mediated liver X receptor-alpha activation inhibits CC chemokine receptor-7 expression on dendritic cells and dampens antitumor responses. *Nat Med* **16**, 98-105 (2010).
19. Baird, J.R., et al. Radiotherapy Combined with Novel STING-Targeting Oligonucleotides Results in Regression of Established Tumors. *Cancer Res* **76**, 50-61 (2016).
20. Deng, L., et al. STING-Dependent Cytosolic DNA Sensing Promotes Radiation-Induced Type I Interferon-Dependent Antitumor Immunity in Immunogenic Tumors. *Immunity* **41**, 843-852 (2014).
21. Harding, S.M., et al. Mitotic progression following DNA damage enables pattern recognition within micronuclei. *Nature* **548**, 466-470 (2017).
22. Crittenden, M.R., et al. Tumor cure by radiation therapy and checkpoint inhibitors depends on pre-existing immunity. *Scientific reports* **8**, 7012 (2018).
23. Lin, J.H., et al. Type 1 conventional dendritic cells are systemically dysregulated early in pancreatic carcinogenesis. *J Exp Med* **217**(2020).
24. Wisdom, A.J., et al. Single cell analysis reveals distinct immune landscapes in transplant and primary sarcomas that determine response or resistance to immunotherapy. *Nature communications* **11**, 6410 (2020).
25. Blair, T.C., Alice, A.F., Zebertavage, L., Crittenden, M.R. & Gough, M.J. The Dynamic Entropy of Tumor Immune Infiltrates: The Impact of Recirculation, Antigen-Specific Interactions, and Retention on T Cells in Tumors. *Front Oncol* **11**, 653625 (2021).

April 7, 2022

RE: Life Science Alliance Manuscript #LSA-2021-01337-TR

Dr. Michael J Gough
Providence Portland Medical Center
Earle A Chiles Research Institute
4805 NE Glisan St
Portland, OR 97213

Dear Dr. Gough,

Thank you for submitting your revised manuscript entitled "Fluorescent tracking identifies key migratory dendritic cells in the lymph node after radiotherapy". We would be happy to publish your paper in Life Science Alliance pending final revisions necessary to meet our formatting guidelines.

- please address the final Reviewer 1 and 3 comments
- please consult our manuscript preparation guidelines <https://www.life-science-alliance.org/manuscript-prep> and make sure your manuscript sections are in the correct order
- please add the author contributions to the main manuscript text
- please use the [10 author names, et al.] format in your references (i.e. limit the author names to the first 10)
- please take a look at your legend for Supplementary Figure 4 and make sure the panels are correct in the text
- please check your figure callouts; you have a callout for Figure 3F, but this is not in the figure or the legend
- please add a callout to Figure 4G in the main manuscript text
- please take a look at your Supplementary Figure 10 and edit accordingly; the figure legend does not have Panels A & B, but the Figure itself has Panels A & B
- please remove the Panel A in your Figure S1 from the legend, the figure, and the figure callout in the main manuscript text; it does not need Panel A because it is the only Panel
- please add a data availability section including the single cell RNA seq data deposition

A. FINAL FILES:

B. MANUSCRIPT ORGANIZATION AND FORMATTING:

Sincerely,

Reviewer #1 (Comments to the Authors (Required)):

In their revised version, Blair et al provided additional evidence in support of their observation strengthening their original observation. I feel that most of the points of my original review were addressed adequately, but a few of the answers remain blurry:

- Please note that the gating strategy presented in S1 does not include the exclusion of F4/80+ so LC are included in the current cDC2 gate provided.
- Pathway analysis 4F remains weak, and lacks statistical evidence

Having said I do not think these issues impede the overall message of the manuscript and I believe that the additional experiments provided by the authors have increased the quality of the manuscript.

Reviewer #2 (Comments to the Authors (Required)):

The authors have addressed all my concerns. I recommend for acceptance.

Reviewer #3 (Comments to the Authors (Required)):

The modifications of this revised manuscript following initial review were substantial and have strengthened the paper substantially. The authors have adequately addressed my initial critiques and have included several essential components in this version of the manuscript such as a characterization of the Kaede model and a more nuanced interpretation of the results. The main limitation to the impact of the findings presented in this manuscript remains the experimental design only considering a few transplanted tumor models. Yet, it still provides important insights into how radiotherapy can directly alter DC trafficking and maturation. Furthermore, in the discussion of this version of the manuscript, the authors have now addressed this limitation. I thank the authors for being responsive to my critiques.

Minor Comments:

1. Panel 4G is not presented in the text. Instead 4F is referred to twice. Please check if the second 4F in the text should instead

be 4G.

-please address the final Reviewer 1 and 3 comments

I believe the only actionable comment is for reviewer 3, as follows:

"1. Panel 4G is not presented in the text. Instead 4F is referred to twice. Please check if the second 4F in the text should instead be 4G. "

In line with the request below, we have modified the text to appropriately refer to the subfigure accordingly.

The reviewers retain some valid reservations on the manuscript, but there are no further actionable items that we can address in this response.

-please consult our manuscript preparation guidelines <https://www.life-science-alliance.org/manuscript-prep> and make sure your manuscript sections are in the correct order

We have corrected the order of sections according to the guidelines

-please add the author contributions to the main manuscript text

We have added an author contributions section using the CREDIT system.

-please use the [10 author names, et al.] format in your references (i.e. limit the author names to the first 10)

This has been corrected all authors if less than 10, and to first 10 if more than or equal to 10.

-please take a look at your legend for Supplementary Figure 4 and make sure the panels are correct in the text

The legend has been corrected to refer to 4C rather than 4E (which does not exist).

-please check your figure callouts; you have a callout for Figure 3F, but this is not in the figure or the legend

This has been corrected to a text reference to Figure 3D. You are correct there is no figure 3F.

-please add a callout to Figure 4G in the main manuscript text

This has been corrected, Figure 4G is now referenced in the text.

-please take a look at your Supplementary Figure 10 and edit accordingly; the figure legend does not have Panels A & B, but the Figure itself has Panels A & B

The legend has been corrected to refer to A) total number of converted mig CD11b+ DCs, B) total number of converted mig CD103+ DCs.

-please remove the Panel A in your Figure S1 from the legend, the figure, and the figure callout in the main manuscript text; it does not need Panel A because it is the only Panel

A) Has been removed from the Figure, legend, and text.

- please add a data availability section including the single cell RNA seq data deposition

This has been added, the data is uploaded, accepted and will be released on April 24th 2022.

April 20, 2022

RE: Life Science Alliance Manuscript #LSA-2021-01337-TRR

Dr. Michael J Gough
Providence Portland Medical Center
Earle A Chiles Research Institute
4805 NE Glisan St
Portland, OR 97213

Dear Dr. Gough,

Thank you for submitting your Research Article entitled "Fluorescent tracking identifies key migratory dendritic cells in the lymph node after radiotherapy". It is a pleasure to let you know that your manuscript is now accepted for publication in Life Science Alliance. Congratulations on this interesting work.

DISTRIBUTION OF MATERIALS:

Again, congratulations on a very nice paper. I hope you found the review process to be constructive and are pleased with how the manuscript was handled editorially. We look forward to future exciting submissions from your lab.

Sincerely,
